

# The double smeared null energy condition

Jackson R. Fliss[1*], Ben Freivogel[1,2†] and Eleni-Alexandra Kontou[1,2‡]

**1** ITFA, Universiteit van Amsterdam, Science Park 904,
1098 XH Amsterdam, the Netherlands
**2** GRAPPA, Universiteit van Amsterdam, Science Park 904,
1098 XH Amsterdam, the Netherlands

⋆ j.r.fliss@uva.nl , † b.w.freivogel@uva.nl , ‡ e.a.kontou@uva.nl

## Abstract

The null energy condition (NEC), an important assumption of the Penrose singularity theorem, is violated by quantum fields. The natural generalization of the NEC in quantum field theory, the renormalized null energy averaged over a finite null segment, is known to be unbounded from below. Here, we propose an alternative, the double smeared null energy condition (DSNEC), stating that the null energy smeared over two null directions has a finite lower bound. We rigorously derive DSNEC from general world-volume bounds for free quantum fields in Minkowski spacetime. Our method allows for future systematic inclusion of curvature corrections. As a further application of the techniques we develop, we prove additional lower bounds on the expectation values of various operators such as conserved higher spin currents. DSNEC provides a natural starting point for proving singularity theorems in semi-classical gravity.



# 1 Introduction

The Null Energy Condition (NEC) is obeyed by all sensible classical theories, but even the most familiar quantum field theories can violate this condition. This violation suggests the possible construction of exotic geometries such as traversable wormholes and bouncing cosmologies in semi-classical gravity. Therefore it is interesting and relevant to investigate (i) the extent of NEC violation that is possible in quantum field theory and (ii) what this implies for physically realizable geometries in semi-classical gravity.

In this paper, we report progress on the first of these fronts. In particular we prove bounds on 'smeared' null energy: the null components of the stress tensor, averaged over a spacetime region. Our main result can be stated roughly as follows. Define the operator $T_{--}^{\text{smear}}$ by averaging over a distance $\delta_+$ in the $x^+$ direction and $\delta_-$ in the $x^-$ direction. We will sometimes refer to this operator the double (null) smeared null energy, or "DSNE". In dimensions higher than 2 this operator is smeared over only a subspace of the entire spacetime.

In the simple context of free scalar quantum fields in Minkowski spacetime, we prove that (schematically)

$$\left\langle T_{--}^{\text{smear}} \right\rangle \geq -\frac{\mathcal{N}_2[\gamma]}{(\delta^+)^{n/2-1}(\delta^-)^{n/2+1}}, \tag{1}$$

where $n$ is the spacetime dimension and $\mathcal{N}_2$ is a dimensionless parameter depending on the number of scalar fields and the details of how the operator is smeared. For massless fields, $\mathcal{N}_2$ is simply proportional to the number of fields; however, for massive fields $\mathcal{N}_2$ depends on the smearing lengths through the dimensionless combination of the mass and the smearing lengths, $\gamma = \delta^+ \delta^- m^2$. For small $\gamma$ and smooth smearing functions, $\mathcal{N}_2$ is an $O(1)$ factor times the number of fields. However, in [1] it was shown that for large masses and in a class of squeezed states $\mathcal{N}_2$ becomes exponentially small in $\gamma$. Here we show, for general states in free theories, that $\mathcal{N}_2 \to 0$ as $\gamma \to \infty$. The precise form of our bound is given in equation (50), and the connection to the schematic form above is demonstrated in (52) and (53).

The universal, power-law dependence on the smearing lengths $\delta^{\pm}$ follows from symmetry arguments, namely the transformation of $T_{--}$ under boosts, as well as the overall engineering dimension of the operator. The nontrivial result is that this operator is indeed bounded from below. This result was first suggested and coined the "Double Smeared Energy Condition" or "DSNEC" in [1] but was not proven there.

Our result is closely related to the Smeared Null Energy Condition (SNEC) [2] which constrains the null energy, averaged over a portion of a single null geodesic. The SNEC, however, does not have a finite field theory limit as its lower bound diverges when the UV cutoff of

the theory goes to zero. Here, we will see that smearing in the perpendicular null direction ("fattening" the geodesic slightly) leads to an operator that is bounded below in quantum field theory. Additionally, we show that the Averaged Null Energy Condition (ANEC) can be derived from DNEC at the appropriate limit for $\delta_- \to \infty$.

Along the way we develop technology for constructing lower bounds on a wide class of smeared operators, at least in the context of free scalars on Minkowski spacetime. As a "test drive" of this technology we also prove smeared bounds on $\phi^2$ expectation values as well as on expectation values of higher-spin currents, $\mathcal{J}_{--...-}$. For the latter we arrive at a lower bound morally similar to (1)

$$\left\langle \mathcal{J}^{\text{smear}}_{--...-} \right\rangle \geq -\frac{\mathcal{N}_{\mathfrak{s}}}{(\delta^+)^{n/2-1}(\delta^-)^{n/2-1+\mathfrak{s}}}, \tag{2}$$

with dependence on the smearing lengths, $\delta^\pm$, fixed by engineering dimension and the transformation of $\mathcal{J}_{--...-}$ under boosts. Here $\mathfrak{s} \in 2\mathbb{N}$ is the spin. This bound implies the "higher-spin ANEC" [3], as we will show later in the paper.

In regards to the relevance of these bounds in semi-classical gravity it is necessary for us to move away from Minkowski space. While we do not attempt to implement the DSNEC in a full semi-classical setting, as a first step we will rephrase our bound in the context of an *absolute inequality* that does not make use of a reference state (such as the Minkowski vacuum). In particular we show that a general world-volume inequality proven by Fewster and Smith [4] implies the DSNEC for massless fields in Minkowski space. This world-volume inequality provides a blueprint for the future incorporation of curvature effects.

All of the above bounds will make use of a fixed momentum space reference frame. As a final result of this paper we will show how to use this ambiguity to our advantage by varying the bound over choices of reference frame. To be specific about the scope of this optimization, we vary over boosts acting on the domain of positive frequencies. The result of this optimization is a lower bound with restored Lorentz covariance and with unexpected, non-linear, dependence on the smearing functions, seen in equation (129). We will show that in even dimensions we can cast this bound in a simple (though still non-linear) form in position space (138).

A brief summary of the organization of this paper is as follows. Below we remark on previous work in the realm of null energy bounds in quantum field theory and will also fix our conventions. In section 2 we discuss renormalization schemes in quantum field theory, including normal ordering (in the context of Minkowski space) and Hadamard renormalization. In section 3 we discuss bounding smeared operators in Minkowski space; in this section we will derive the precise form of the DSNEC (section 3.2), bounds on $\phi^2$ (section 3.3), and on higher-spin currents (section 3.4). Afterwards, in section 4, we will recast the DSNEC in the context of an absolute quantum energy inequality. After a brief introduction of the relevant technology (section 4.1) we will rederive the massless DSNEC and discuss massive corrections to the bound (section 4.3). Following that, in section 5 we perform the optimization over boosted domains and discuss the form of the lower bound we find. Lastly, in the discussion, section 6, we will discuss our results in the context of field theory and in semi-classical gravity and what open questions remain at this stage.

**Relation to previous work.**

Ford [5] was the first to introduce bounds on the averaged renormalized energy density and flux of quantum fields now known as Quantum Energy Inequalities (QEIs). QEIs generally express restrictions in duration and magnitude of negative energies in the context of quantum field theory.

Since then, there has been much progress proving QEIs for a variety of fields on flat and curved spacetimes (see [6] and [7] for recent reviews). Most of these results are for averages

over timelike curves. Here, we focus on progress on bounds over null geodesics.

- **ANEC:** The averaged null energy condition (ANEC) states that the integral of the null energy (classical or quantum) over an entire null (achronal) geodesic is non-negative

$$\int_{-\infty}^{\infty} d\lambda \, \langle T_{--} \rangle \geq 0 \,. \tag{3}$$

The ANEC has been proven for flat [8] and curved spacetimes [9] for free fields using QEIs and in Minkowski spacetime for interacting fields using general quantum information bounds [10] and causality [3]. It also follows from the quantum null energy condition (QNEC) [11], discussed below, using holography [12], and from the monotonicity of relative entropy [13], just to mention a few results. In section 3.2 we show that the ANEC follows from the DSNEC. There are no known counterexamples to self-consistent achronal ANEC in the semi-classical regime.

- **Null QEIs:** The first null QEIs bounds were obtained in two spacetime dimensions, starting with Flanagan [14] for free fields in flat and curved spacetimes. Fewster and Hollands [15] proved a null QEI for classes of interacting conformal field theories (CFTs), a result recently generalized to curved spacetimes [16].

In four spacetime dimensions the situation is very different. Fewster and Roman [17] showed using an explicit counterexample that finite lower bounds of null QEIs do not exist. In their work they used a sequence of vacuum–plus–two–particle states. As the three-momenta of the excited modes become more and more parallel to the spatial part of the null vector tangent to the geodesic, the bound diverges to negative infinity.

To circumvent that problem, Freivogel and Krommydas [2] suggested the SNEC

$$\int_{-\infty}^{+\infty} d\lambda \, g^2(\lambda) \langle T_{--} \rangle \geq -\frac{4B}{G_N} \int_{-\infty}^{+\infty} d\lambda \left( g'(\lambda) \right)^2 \,, \tag{4}$$

where $g(\lambda)$ is a differentiable 'smearing function' that controls the region where the null energy is averaged, $B$ is an unknown dimensionless constant and $G_N$ is the Newton constant. When gravity is coupled to such theories, the renormalized $G_N$ to 1-loop order is $G_N \sim \ell_{UV}^{n-2}/N$ where $\ell_{UV}$ is the UV cutoff of the theory and $N$ the number of fields. The presence of the UV cutoff ensures that the bound remains finite in cases such as the Fewster-Roman counterexample. The SNEC has been proven for free fields in Minkowski spacetime [1] but such a proof cannot easily be generalized for interacting fields and spacetimes with curvature. Additionally the bound diverges when the UV cutoff is taken to zero. We comment on the relationship between DSNEC and SNEC in Appendix B.

- **QNEC:** The Quantum Null Energy Condition (QNEC) [18] is an extension of the NEC to a local lower bound on the null stress tensor valid in generic quantum field theories in Minkowski space-time [11,19,20]. The QNEC bounds a state's null energy at a point by the second variation of the entanglement entropy of the state reduced on a portion of a null hypersurface with respect to infinitesimal null-deformations of its entangling surface:

$$\langle T_{--} \rangle \geq \frac{1}{2\pi a} S_{\text{ent}}'' \,, \tag{5}$$

where $a$ is the induced area element on the entangling surface at the given point. In this sense, the QNEC is a *state dependent* bound of an entirely different character than discussed in section 2 since the right-hand side cannot be written as the expectation value of an operator.

- **Singularity theorems:** The Penrose singularity theorem [21] proves null geodesic incompleteness using as an assumption the NEC thus it is inapplicable in semiclassical gravity. Efforts to weaken the energy condition required in the theorem started with the works of Tipler [22] and Borde [23]. Fewster and Galloway [24] and more recently Fewster and Kontou [25] proved singularity theorems with conditions inspired by QEIs. The first semiclassical singularity theorem for timelike geodesic incompleteness was recently proven [26] and the required initial contraction estimated for cosmological spacetimes. An analogous theorem for null geodesic incompleteness was proven using SNEC as an assumption [16]. While we do not prove a singularity theorem in this work, the derivation of DSNEC is partly inspired by the need to have a semiclassical replacement of the NEC as an assumption to singularity theorems.

- **QFC:** A different approach is the Quantum Focusing Conjecture (QFC) [18] which provides an elegant proposal for how to generalize singularity theorems to semi-classical gravity by promoting the classical expansion to a *quantum expansion*. It depends both on null variations of the geometric area element and the outer entanglement entropy. Thus it has a state-dependence of a similar character to the QNEC and in fact the QNEC follows as a consequence of the QFC. This poses the following issue with the QFC: the initial condition needed to prove a singularity theorem is the non-positive quantum expansion on some surface. However, it is not clear that the quantum expansion is an observable quantity (again, since the entanglement entropy cannot be written as the expectation value of an operator). Our approach is complementary to the QFC and appropriate for proving singularity theorems from purely geometric aspects of the initial surface.

## Conventions

Unless otherwise specified, we work in $n$ spacetime dimensions, assume $\hbar = c = 1$ and use metric signature $(+, -, \ldots, -)$. While we will make statements involving general metrics, $g_{\mu\nu}$, we will perform concrete calculations primarily in Minkowski space. When considering null subspaces we will denote, w.l.o.g., null coordinates[1] $x^{\pm} = t \pm x^1$ and transverse coordinates, $\vec{y} = (x^2, \ldots, x^n)$:

$$ds^2 = dt^2 - \sum_{i=1}^{n} (dx^i)^2 = dx^+ dx^- - \sum_{a=2}^{n} (dy^a)^2. \tag{6}$$

Null derivatives will be denoted as $\partial_{\pm} := \frac{1}{2}(\partial_t \pm \partial_1)$. In momentum space this implies the following notation $k_{\pm} := \frac{1}{2}(k_0 \pm k_1)$; the inner product with coordinates remains unchanged, $k_{\mu} x^{\mu} = k_0 t + k_i x^i = k_+ x^+ + k_- x^- + k_a y^a$.

For the Fourier transform we use the following convention

$$\tilde{f}(k) = \int_{\mathbb{R}^n} d^n x f(x) e^{ikx} = \int dt \, dx^1 \, d^{n-2} \vec{y} \, f(x) e^{ikx}. \tag{7}$$

Finally, for future reference, we define $f_{\sqrt{G}}$, the square-root of a normalized Gaussian with unit variance:

$$f_{\sqrt{G}}(s) := \frac{1}{(2\pi)^{1/4}} e^{-s^2/4}. \tag{8}$$

---

[1]Note importantly a discrepancy in integration measures $dt \, dx^1 = \frac{1}{2} dx^+ dx^-$. In the interest of comparison to previous results and to be clear on this front, we will always denote integrations with respect to null-coordinates by $d^2 x^{\pm} := dx^+ dx^-$. Similarly integrations in momentum space will follow a similar notation: $d^2 k_{\pm} := dk_+ dk_- = \frac{1}{2} dk_0 dk_1$.

## 2 Renormalization in quantum field theory

We consider the massive minimally coupled classical scalar field $\phi$ with field equation

$$(\Box_g + m^2)\phi = 0,\tag{9}$$

where $m$ has dimensions of inverse length. The Lagrangian is

$$L[\phi] = \frac{1}{2}\left((\nabla\phi)^2 - m^2\phi^2\right).\tag{10}$$

Varying the action with respect to the metric gives the stress-energy tensor

$$T_{\mu\nu} = \nabla_\mu\phi\nabla_\nu\phi - \frac{1}{2}g_{\mu\nu}g^{\lambda\rho}\nabla_\lambda\phi\nabla_\rho\phi + \frac{1}{2}m^2 g_{\mu\nu}.\tag{11}$$

After quantization, our main object of interest is the two point function,

$$W_\psi(x,x') \equiv \langle\phi(x)\phi(x')\rangle_\psi,\tag{12}$$

where $\psi$ is a quantum state of interest. The class of states we consider in this paper are the *Hadamard states* [27] whose two point-functions have well-known singularity structures.

We renormalize the stress-energy tensor following the prescription of Hollands and Wald [28, 29]. In these works they present the axioms that express the desired properties of local time-ordered products of fields. Those properties include locality, continuity, analyticity, symmetry of the factors and unitarity. The procedure described below defines a renormalized stress-tensor that obeys these properties up to finite renormalization freedoms.

First let's define the point-split stress-energy operator

$$\mathbb{T}^{\text{split}}_{\mu\nu'}(x,x') = \nabla^{(x)}_\mu \otimes \nabla^{(x')}_{\nu'} - \frac{1}{2}g_{\mu\nu'}(x,x')g^{\lambda\rho'}(x,x')\nabla^{(x)}_\lambda \otimes \nabla^{(x')}_{\rho'} + \frac{1}{2}m^2 g_{\mu\nu'}(x,x')\mathbb{1}\otimes\mathbb{1},\tag{13}$$

where $g_{\mu\nu'}(x,x') = g_{\mu\rho}(x)g^\rho{}_{\nu'}(x,x')$ is the parallel propagator implementing parallel transport of vectors along the unique geodesic connecting $x$ and $x'$ (we assume the points are close enough to be in a geodesic convex neighborhood). That is, if $V$ is a tangent vector on $x'$, then the vector at $x$ after parallel transport along the geodesic is given by

$$V^\mu(x) = g^\mu{}_{\nu'}(x,x')V^{\nu'}(x'),\tag{14}$$

which defines $g^\mu{}_{\nu'}$. Note that in the coincidence limit

$$\lim_{x\to x'} g^\mu{}_{\nu'}(x,x') = g^\mu{}_\nu(x') = \delta^\mu_\nu.\tag{15}$$

Then we can define

$$\langle T^{\text{fin}}_{\mu\nu}\rangle_\psi(x) = \lim_{x'\to x} g_\nu{}^{\nu'}(x,x')\mathbb{T}^{\text{split}}_{\mu\nu'} \circ (W_\psi - H_{(k)})(x,x'),\tag{16}$$

where "$\circ$" denotes the action of the differential operator $\mathbb{T}^{\text{split}}$ on the bi-distribution $W_\psi - H_{(k)}$. $H_{(k)}$ are terms up to order $k$ of the *Hadamard parametrix*, a bi-distribution that encodes the singularity structure of the two-point function of Hadamard states, expressed as an infinite series [30]. As an example to keep in mind, for the massless free scalar the Hadamard parametrix coincides with the Minkowski vacuum two-point function. We are being schematic now but we will discuss this parametrix in more detail in section 4 where its details are more relevant.

If $\psi$ is Hadamard then any open globally hyperbolic subset of the manifold is also considered as a spacetime. Then we have $W_\psi - H_{(k)} \in C^2$ for $k$ large enough in any globally

hyperbolic convex normal neighbourhood. We should note that every point $x$ on the manifold has such a neighbourhood called an ultra-regular domain [4].

Then $\mathbb{T}^{\text{split}}_{\mu\nu'} \circ (W - H_{(k)})(x, x')$ is defined and continuous along coincident points. As a bit of short-hand for the future, we will denote the coincident limit of a generic bi-distribution, $\mathcal{B}(x, x')$, as

$$[\![\mathcal{B}]\!](x') = \lim_{x \to x'} \mathcal{B}(x, x'). \tag{17}$$

By $\langle T^{\text{ren}}_{\mu\nu} \rangle$ we denote the expectation value of the renormalized stress-energy tensor following the axioms of [28, 29]. The difference of $\langle T^{\text{ren}}_{\mu\nu} \rangle$ between two Hadamard states $\psi$ and $\psi_0$ is smooth at the coincident limit $x \to x'$

$$\langle T^{\text{ren}}_{\mu\nu} \rangle_\psi - \langle T^{\text{ren}}_{\mu\nu} \rangle_{\psi_0} = [\![ g_\nu{}^{\nu'} \mathbb{T}^{\text{split}}_{\mu\nu'} \circ (W_\psi - W_{\psi_0}) ]\!], \tag{18}$$

where

$$\langle T^{\text{ren}}_{\mu\nu} \rangle_\psi(x) = \langle T^{\text{fin}}_{\mu\nu} \rangle_\psi - Q(x)g_{\mu\nu}(x) + C_{\mu\nu}(x). \tag{19}$$

The definition includes the remaining finite renormalization freedom which takes the form of a state-independent conserved local curvature term $C_{\mu\nu}$ that vanishes in Minkowski space. Here $Q$ is a term introduced by Wald [31] to preserve the conservation of the stress-energy tensor.

In Minkowski space we have a distinguished state, the Minkowski vacuum, annihilated by the generators of the Poincaré group. We will always denote this state by $\Omega$. This defines a canonical renormalization scheme via subtraction by the Minkowski vacuum, i.e. *normal ordering*. We will denote it by $:\ :$ as is customary

$$\langle :T_{\mu\nu}: \rangle_\psi := [\![ g_\nu{}^{\nu'} \mathbb{T}^{\text{split}}_{\mu\nu'} \circ (W_\psi - W_\Omega) ]\!]. \tag{20}$$

We will extend this definition to a general operator statement:

$$:\mathcal{O}: \equiv \mathcal{O} - \langle \mathcal{O} \rangle_\Omega. \tag{21}$$

The Hadamard series coincides with the singularity structure of the Minkowski vacuum and so in Minkowski space

$$\langle :T_{\mu\nu}: \rangle_\psi = \langle T^{\text{fin}}_{\mu\nu} \rangle_\psi \tag{22}$$

as local operators. Additionally $:T_{--}:$ coincides with $T^{\text{ren}}_{--}$ in Minkowski space since terms proportional to the metric are killed by contraction with null vectors (these operators do not have to coincide however for the energy density). When it is clear by context that we are working in Minkowski space (for example in the next section) we will drop the $:\ :$ from $T_{\mu\nu}$ with it being clear the renormalization scheme being used.

Because of the subtraction of divergences, operators that are classically positive can acquire negative quantum expectation values after renormalization. It is the goal of this paper to diagnose the magnitude of this negative expectation value in the form of a lower bound, or a quantum inequality. The most general form of a quantum inequality that bounds $\mathcal{O}$ is

$$\langle \mathcal{O}(f) \rangle_\Psi \geq -\langle \mathcal{Q}(f) \rangle_\Psi, \tag{23}$$

where $f$ is a non-negative smearing function on spacetime. In general the operator $\mathcal{Q}(f)$ could be an unbounded operator.

We call *difference* QEIs the ones where we bound the smooth difference between the expectation values of two Hadamard states

$$\langle \mathcal{O}(f) \rangle_\Psi - \langle \mathcal{O}(f) \rangle_{\Psi_0} \geq -\langle \mathcal{Q}_{\Psi_0}(f) \rangle_\Psi. \tag{24}$$

In that case the bound can depend on both the reference state $\Psi_0$ and the state of interest $\Psi$. If instead we renormalize using the Hadamard parametrix the QEI is called *absolute*. If the reference state is the massless Minkowski vacuum the two kinds coincide. If the bound depends on $\Psi$ then it is a *state dependent* bound. The bounds of interest in this paper are *state independent* and take the form

$$\langle \mathcal{O}(f) \rangle_\Psi - \langle \mathcal{O}(f) \rangle_{\Psi_0} \geq -\mathcal{Q}_{\Psi_0}(f) \,. \tag{25}$$

## 3 Derivation of a general Minkowski bound

In this section we derive a quantum energy inequality for Minkowski spacetime over a general domain. We want to bound smeared quantities of the form

$$A_{\mathcal{O}\mathcal{O}} \equiv \int_{\Sigma_p} d^p x \, g(x)^2 \langle : \mathcal{O}(x)^2 : \rangle_\psi \,, \tag{26}$$

where $\Sigma_p$ is a $p$-dimensional time-like subspace of $\mathbb{R}^{1,n-1}$. We will only consider time-like subspaces in this paper as it is expected that if $\Sigma_p$ is a space-like subspace then the right hand side of any prospective bound will diverge leaving the bound trivial. We will additionally assume from here on that $\Sigma_p$ is flat and translationally invariant such that fields admit a (partial) Fourier transform along $\Sigma_p$ and denote the space of these momenta as $\tilde{\Sigma}_p$. Using this partial Fourier transform we can fictitiously "point-split" the operators:

$$A_{\mathcal{O}\mathcal{O}} = \int_{\tilde{\Sigma}_p} \frac{d^p \xi}{(2\pi)^p} \int_{\Sigma_p} d^p x \, d^p x' \, e^{i\xi \cdot (x-x')} g(x) g(x') \big( \langle \mathcal{O}(x) \mathcal{O}(x') \rangle_\psi - \langle \mathcal{O}(x) \mathcal{O}(x') \rangle_\Omega \big) \,. \tag{27}$$

We emphasize that at this point both terms in (27) possess contact singularities: indeed (27) is still exactly equal to (26) as the momentum integration simply induces a delta-function. It is the difference of the two terms that is finite. Now we make the main assumption

**Assumption 1**: *The commutator of $\mathcal{O}$ with itself is a c-number:*

$$[\mathcal{O}(t,\vec{x}), \mathcal{O}(0,\vec{0})] \propto \mathbb{1} \,, \tag{28}$$

*where $\mathbb{1}$ is the identity operator on the Hilbert space.*

This assumption is certainly satisfied when $\mathcal{O}$ is a free field or derivative there-of. More generally Assumption 1 is very constraining and as explained in Appendix A, it is likely that this assumption is only satisfied by *generalized free fields*, i.e. operators whose higher-point functions can be evaluated via Wick contractions.[2] We pause to note that while for most of this paper we will focus on free theories, the construction in this section and the bound (36) are valid e.g. for interacting theories with some large $N$ parameter that suppresses non-Wick terms in $\mathcal{O}$ higher-point functions by powers of $1/N$, in the $N \to \infty$ limit. As a word of caution, however, the bounds we are able to derive for generalized free field theories apply to operators that are quadratic in the generalized free field itself or derivatives of the generalized free field. This does *not* include the stress tensor unless the field is exactly free.

Under Assumption 1 the difference appearing in the integrand of (27) is symmetric under $x \leftrightarrow x'$ and we can then restrict the $k$ integration to a half-space,[3] $D \subset \tilde{\Sigma}_p$. Importantly, the

---

[2]We thank Tarek Anous and Mert Besken for a discussion on this point.

[3]That is, under the parity map $\mathbf{P} : \vec{k} \to -\vec{k}$ on $\tilde{\Sigma}_p$, $D$ is such that $\tilde{\Sigma}_p = D \sqcup \mathbf{P}(D)$.

first term of (27) is a positive-definite regardless of the choice of domain as it can be written as the inner product

$$\int_D \frac{d^p\xi}{(2\pi)^p} \langle \mathcal{O}_g(\xi)\psi | \mathcal{O}_g(\xi)\psi \rangle, \qquad |\mathcal{O}_g(\xi)\psi\rangle := \int d^p x e^{i\xi x} g(x)\mathcal{O}(x)|\psi\rangle. \qquad (29)$$

Thus if we are interested in bounding $A_\mathcal{O}$ from below, it suffices to focus on the second term

$$A_{\mathcal{O}\mathcal{O}} \geq -\mathcal{Q}_{\mathcal{O}\mathcal{O}}[D], \quad \mathcal{Q}_{\mathcal{O}\mathcal{O}}[D] \equiv 2\int_D \frac{d^p\xi}{(2\pi)^p} \int_{\Sigma_p} d^p x d^p x' \, g(x)g(x')e^{i\xi(x-x')}\langle\mathcal{O}(x)\mathcal{O}(x')\rangle_\Omega. \qquad (30)$$

It is worth clarifying the previous step, the role of $D$ in (30), and why (30) is finite when both terms in (27) were individually divergent: because of the structure of divergences of Hadamard states any potential contact divergences are exactly cancelled in the difference in (27). To this finite expression we restrict the momentum integration to $D$; both terms in the integrand are now finite and we drop the obviously positive terms. As a result of restriction of momentum integration to $D$, the $\xi$ integral is now prohibited from reproducing a delta function $\delta^p(x - x')$; this changes the "fictitious point splitting" in (27) to an effective point splitting, softening the contact divergences in (30) (we will soon see an explicit example illustrating this) and allowing us to write non-trivial lower bounds. After dropping the first, positive, term we are not allowed to reverse the logic and "re-extend" $D$ to a full momentum integration and potentially rediscover contact singularities: the integrand of (30) is no longer symmetric under $x \leftrightarrow x'$ because the commutator of $\mathcal{O}(x)$ and $\mathcal{O}(x')$ is generically not zero. Within the regime of validity of our main assumption, (30) is otherwise fairly general and valid for arbitrary dimensions, masses, etc. If one knows the vacuum 2-point function in position space, one can pick a domain $D$ and just integrate.

Let us illustrate this logic with a simple choice of domain that we call the *canonical domain*; it is given by the half-space of positive frequencies in $\tilde{\Sigma}_p$:

$$D_0 := \{\xi \in \tilde{\Sigma}_p \,|\, \xi_0 \geq 0\}, \qquad (31)$$

for which the bounds take the general form

$$\mathcal{Q}_{\mathcal{O}\mathcal{O}}[D_0] = 2\int_{\Sigma_p} d^p x d^p x' \, g(x)g(x')\frac{i\delta^{p-1}(\vec{x}-\vec{x}')}{t-t'+i\epsilon}\langle\mathcal{O}(x)\mathcal{O}(x')\rangle_\Omega. \qquad (32)$$

As we see, the contact divergence in the two-point function $\langle\mathcal{O}(x)\mathcal{O}(x')\rangle_\Omega$ is softened by the kernel $i(t-t'+i\epsilon)^{-1}$ which effectively point-splits it.

When $p \geq 2$, we also have a family of bounds obtained by boosting $D_0$ in (w.l.o.g) the $(\xi^0, \xi^1)$ plane by a parameter $\eta \in \mathbb{R}$:

$$D_\eta := \{\xi \in \tilde{\Sigma}_p \,|\, \xi_\eta := e^\eta \xi_+ + e^{-\eta}\xi_- \geq 0\}. \qquad (33)$$

It is clear that when $p \geq 2$ that $D_0$ is a special case of $D_\eta$ with $\eta = 0$.

In general a choice of domain $D$ breaks any subgroup, $H_{\tilde{\Sigma}_p} \subset SO(1, n-1)$, of Lorentz invariance originally possessed by $\tilde{\Sigma}_p$ and so, unsurprisingly, (30) will depend on a fixed reference frame. However since (30) applies, at least in principle, for any domain, $D$, we are free to optimize over choices of $D$,

$$A_{\mathcal{O}\mathcal{O}} \geq -\min_D \mathcal{Q}_{\mathcal{O}\mathcal{O}}[D], \qquad (34)$$

and we expect this minimization to restore covariance under $H_{\tilde{\Sigma}_p}$.[4] In practice, however, this is somewhat unwieldy minimization; in this paper we will content ourselves with varying over the much smaller family of boosted domains, $D_\eta$, as this provides a controlled one-parameter minimization over $\eta \in \mathbb{R}$. The minimization over $D_\eta$ will likely not provide the tightest bound, but will restore covariance under boosts in the $(k_0, k_1)$ plane. In general this will not restore the full $H_{\tilde{\Sigma}_p}$ covariance (since this subgroup could consist of boosts in multiple directions plus rotations in the internal space) but for the two-dimensional time-like domains we primarily consider in this paper, this boost minimization will restore covariance.

Generally, the position space correlators are not simple objects to work with (e.g. in massive theories). It will be convenient for us to express the above entirely in the Fourier space, $\tilde{\Sigma}_p$. Translational invariance of the vacuum implies

$$\langle \tilde{\mathcal{O}}(k)\tilde{\mathcal{O}}(k')\rangle_\Omega := (2\pi)^p \delta^p(k+k') G_{\mathcal{O}\mathcal{O}}(k'), \tag{35}$$

for some $G_{\mathcal{O}\mathcal{O}}$. We then obtain

$$\mathcal{Q}_{\mathcal{O}\mathcal{O}}[D] = 2 \int_{\tilde{\Sigma}_p} \frac{d^p k}{(2\pi)^p} |\tilde{g}(k)|^2 \int_D \frac{d^p \xi}{(2\pi)^p} \, G_{\mathcal{O}\mathcal{O}}(k-\xi). \tag{36}$$

Note the difference in integration regions between $\xi$ and $k$. We have written this formula in a convenient and general form; below we apply it to some specific situations.

### 3.1 Two-dimensional smeared null-energy

To begin let's apply the bound (36) to the null-energy of a two-dimensional massive scalar smeared over spacetime (i.e. we will take $\Sigma_2 = \mathbb{R}^{1,1}$). Indeed, $T_{--}$ can be written in the form $:\mathcal{O}\mathcal{O}:$,

$$T_{--}(x^+, x^-) =: \partial_-\phi \partial_-\phi : (x^+, x^-), \tag{37}$$

and so identifying

$$G_{\partial_-\phi\partial_-\phi}(k) = \frac{1}{8} k_- \Theta(k_-)(2\pi)\delta\left(k_+ - \frac{m^2}{4k_-}\right), \tag{38}$$

we have

$$\mathcal{Q}_{T_{--}}[D] = 2 \int_D \frac{d^2\xi_\pm}{(2\pi)^2} \int_0^\infty \frac{d\zeta_-}{(2\pi)} \zeta_- |\tilde{g}(k_\pm)|^2 \Big|_{k_-=\xi_-+\zeta_-, \, k_+=\xi_++\frac{m^2}{4\zeta_-}} . \tag{39}$$

More specifically, we can investigate $\mathcal{Q}_{T_{--}}[D_\eta]$ for the boosted domains, (33):

$$\mathcal{Q}_{T_{--}}[D_\eta] = 2 \int \frac{d^2 k_\pm}{(2\pi)^2} \int_0^\infty \frac{d\zeta_-}{2\pi} |\tilde{g}(k_\pm)|^2 \zeta_- \Theta\left(k_\eta - e^{-\eta}\zeta_- - e^\eta \frac{m^2}{4\zeta_-}\right), \tag{40}$$

where $k_\eta = e^\eta k_+ + e^{-\eta} k_-$. Doing the linear $\zeta_-$ integral between the endpoints of Heaviside domain, $\frac{1}{2}e^\eta\left(k_\eta \pm \sqrt{k_\eta^2 - m^2}\right)$, we find

$$\mathcal{Q}_{T_{--}}[D_\eta] = \frac{e^{2\eta}}{2\pi} \int \frac{d^2 k_\pm}{(2\pi)^2} |\tilde{g}(k_\pm)|^2 k_\eta \sqrt{k_\eta^2 - m^2}\Theta(k_\eta - m), \tag{41}$$

---

[4]The argument is the following: any $D_{min}$ found through a variational principle will be stationary under infinitesimal changes of frame. This includes infinitesimal $H_{\tilde{\Sigma}_p}$ transformations.

leading to a bound[5]

$$\int d^2x^{\pm}\, g(x^{\pm})^2 \langle T_{--}(x^{\pm})\rangle_{\psi}^{(2d)} \geq -\min_{\eta \in \mathbb{R}} \frac{e^{2\eta}}{\pi} \int \frac{d^2k_{\pm}}{(2\pi)^2} |\tilde{g}(k_{\pm})|^2\, k_{\eta} \sqrt{k_{\eta}^2 - m^2}\, \Theta(k_{\eta}-m). \tag{43}$$

## 3.2 Double smeared null energy in higher dimensions

We now look to apply (36) to the null-energy smeared in two null directions in dimensions $n \geq 3$ in what was coined the *DSNE* in [1]. Since we are interested in smearing in $x^{\pm}$ the relevant domain, $\Sigma_2$, is the $(x^+, x^-)$ plane defined by the level-set $\Sigma_2 = \{(x^+, x^-; y^a = 0)|a = 2, \ldots, n-1\}$. The "partial Fourier transform" $\tilde{\Sigma}_p$ is then spanned by two momenta, $k^{\pm}$.

To be specific we will continue to work with a free massive scalar field. The relevant momentum space correlator is

$$\langle \partial_-\phi(k_+, k_-, \vec{y} = 0)\partial_-\phi(k'_+, k'_-, \vec{y} = 0)\rangle_{\Omega} = (2\pi)^2 \delta^2(k+k')G_{T_{--}}(k'), \tag{44}$$

with

$$G_{T_{--}}(k) = \frac{V_{n-3}}{2(2\pi)^{n-3}} k_-^2 (4k_+k_- - m^2)^{\frac{n-4}{2}} \Theta\left(4k_+k_- - m^2\right)\Theta(k_-), \tag{45}$$

where $V_{n-3} = (2\pi^{\frac{n-2}{2}})/\Gamma\left(\frac{n-2}{2}\right)$ is the volume of the angular $S^{n-3}$. The bound on the stress tensor is, for general mass and dimension,

$$\int d^2x^{\pm}\, g(x^{\pm})^2 \langle T_{--}(x^{\pm}, \vec{y} = 0)\rangle_{\psi} \geq -\min_{D} \mathcal{Q}_{T_{--}}[D], \tag{46}$$

where we will write

$$\mathcal{Q}_{T_{--}}[D] = \int \frac{d^2k_{\pm}}{(2\pi)^2} |\tilde{g}(k_{\pm})|^2 h_D(k_{\pm}), \tag{47}$$

so that the function $h_D(k)$ encodes the choice of reference frame:

$$h_D(k) = \frac{8V_{n-3}}{(2\pi)^{n-3}} \int \frac{d^2\xi_{\pm}}{(2\pi)^2} \Theta(\xi_{\pm} \in D)\zeta_-^2 \left(4\zeta_+\zeta_- - m^2\right)^{\frac{n-4}{2}} \Theta\left(4\zeta_+\zeta_- - m^2\right)\Theta(\zeta_-)\Big|_{\zeta_{\pm} = k_{\pm} - \xi_{\pm}}. \tag{48}$$

Processing $h_D$ a bit by changing integration variables from $\xi_{\pm}$ to $\zeta_{\pm}$ we find for the boosted domain, $D_{\eta}$,

$$h_{D_{\eta}}(k_{\pm}) = \frac{8V_{n-3}}{(2\pi)^{n-3}} \int \frac{d^2\zeta_{\pm}}{(2\pi)^2} \zeta_-^2 (4\zeta_+\zeta_- - m^2)^{\frac{n-4}{2}} \Theta(4\zeta_+\zeta_- - m^2)\Theta(\zeta_-)\Theta(k_{\eta} - \zeta_{\eta})$$

$$= \frac{e^{2\eta}}{(4\pi)^{\frac{n-1}{2}}} \frac{1}{\Gamma\left(\frac{n+1}{2}\right)} k_{\eta}(k_{\eta}^2 - m^2)^{\frac{n-1}{2}} \Theta(k_{\eta} - m), \tag{49}$$

where $k_{\eta} = e^{\eta}k_+ + e^{-\eta}k_-$. This provides a "first principles" derivation of the bound suggested in [1] by investigating $T_{++}$ expectation values in squeezed states:

$$\int_{\Sigma_2} d^2x^{\pm}\, g(x^{\pm})^2 \langle T_{--}(x^{\pm}, \vec{y} = 0)\rangle_{\psi} \geq -\min_{\eta \in \mathbb{R}} c_{T_{--}}^{(n)} e^{2\eta} \int \frac{d^2k_{\pm}}{(2\pi)^2} |\tilde{g}(k_{\pm})|^2\, k_{\eta}(k_{\eta}^2 - m^2)^{\frac{n-1}{2}} \Theta(k_{\eta} - m), \tag{50}$$

---

[5]This bound differs from an apparent factor of 4 from that appearing in the appendix of [1] stemming from a difference in normalization of the Fourier transform here (equation (7)) and in [1]:

$$\tilde{g}_{here}(k) := \int d^2x\, e^{ikx}g(x) = \frac{1}{2}\int d^2x^{\pm}\, e^{ikx}g(x) \equiv \frac{1}{2}\tilde{g}_{there}(k). \tag{42}$$

This factor of 4 follows all comparisons to results in [1].

with

$$c_{T_{--}}^{(n)} = \frac{1}{(4\pi)^{\frac{n-1}{2}}\Gamma\left(\frac{n+1}{2}\right)}.$$

(51)

To make the matching to [1] more explicit, let us call $g(x^+, x^-) = \frac{1}{\sqrt{\delta^+\delta^-}}\mathcal{F}(x^+/\delta^+, x^-/\delta^-)$, where $\mathcal{F}(s^+, s^-)$ is a function of dimensionless variables dropping off quickly for $|s^\pm| \gg 1$ and normalized to $\int d^2s^\pm \mathcal{F}(s^+, s^-)^2 = 1$. Calling $e^{\tilde{\eta}} \equiv \sqrt{\frac{\delta^-}{\delta^+}}e^\eta$ and $\rho_\pm \equiv \delta^\pm k_\pm$, and denoting $\rho_{\tilde{\eta}} = e^{\tilde{\eta}}\rho_+ + e^{-\tilde{\eta}}\rho_-$, the universal power-law dependence on $\delta^+$ and $\delta^-$ can be scaled out of the integral:

$$\int_{\Sigma_2} \frac{d^2x^\pm}{\delta^+\delta^-}\mathcal{F}(x^+/\delta^+, x^-/\delta^-)^2 \langle T_{--}(x^\pm, \vec{y} = 0)\rangle_\psi \geq -\frac{\mathcal{N}_{2,n}[\gamma]}{(\delta^+)^{\frac{n-2}{2}}(\delta^-)^{\frac{n+2}{2}}},$$

(52)

where

$$\mathcal{N}_{2,n} := \min_{\tilde{\eta}\in\mathbb{R}} c_{T_{--}}^{(n)} e^{2\tilde{\eta}} \int \frac{d^2\rho_\pm}{(2\pi)^2}|\tilde{\mathcal{F}}(\rho_+, \rho_-)|^2 \rho_{\tilde{\eta}}(\rho_{\tilde{\eta}}^2 - \gamma^2)^{\frac{n-1}{2}}\Theta(\rho_{\tilde{\eta}} - \gamma)$$

(53)

is now a dimensionless parameter depending on the (dimensionless) Fourier-transformed smearing function, $\tilde{\mathcal{F}}(\rho_\pm) := \int d^2s\, e^{i\rho_\pm s^\pm}\mathcal{F}(s^\pm)$ and the dimensionless combination of the mass and smearing lengths, $\gamma^2 := \delta^+\delta^- m^2$. This is precisely the form of the bound proposed in the introduction, (1), and what was referred to as the DSNEC in [1]. Note that in [1] $\tilde{\eta}$ was implicitly set to zero however we have left it as tuneable degree of freedom in our bound above. This suggests, following the discussion at the beginning of this section, a further optimization over $\tilde{\eta}$. We will do this in section 5.

Equation (53) shows that the prefactor $\mathcal{N}_{2,n} \to 0$ as the mass becomes large compared to the smearing lengths. To see this, note that as $\gamma \to \infty$, the theta function restricts the integral on the right side of (53) to very large $\rho_{\tilde{\eta}}$. (For this discussion, we can just pick any value of the boost $\tilde{\eta}$.) Since the smearing function $\tilde{\mathcal{F}}(\rho_\pm)$ must fall off at large dimensionless momenta $\rho$, the integrand becomes small in the region of integration, so the entire expression approaches zero as $\gamma \to \infty$.

**ANEC**

Having derived the DSNEC we now take a brief opportunity to show that it implies the ANEC. We want to take the limit $\delta^+ \to 0$ and $\delta^- \to \infty$ while holding $\delta^+\delta^- \equiv \alpha^2$ fixed. To recover the ANEC limit we require that the smearing function satisfies

$$\lim_{\delta^+\to 0}\lim_{\delta^-\to\infty}\frac{1}{\delta^+}\mathcal{F}(x^+/\delta^+, x^-/\delta^-)^2 = A\delta(x^+ - \beta),$$

(54)

where $A$ and $\beta$ are real numbers. An example of such a function that satisfies (54) is the Gaussian, $\mathcal{F}(s^+, s^-) = f_{\sqrt{G}}(s^+)f_{\sqrt{G}}(s^-)$. Then Eq.(52) becomes

$$\int_{-\infty}^{\infty} dx^- \langle T_{--}(x^+ = \beta, x^-, \vec{y} = 0)\rangle_\psi \geq -\lim_{\delta^+\to 0}\frac{\mathcal{N}_{2,n}[\gamma]}{A\alpha^n}\delta^+ = 0.$$

(55)

We note that $\mathcal{N}_{2,n}$ remains fixed in this limit.

## 3.3 The smeared $\phi\phi$ correlator

Though the main focus of this paper is on smeared null-energy, we comment on the generality of (30) by applying to two additional situations in this section and in section 3.4. To start, we

can posit a bound on the smeared correlator of the $n$-dimensional massive scalar smeared over two null dimensions, $\Sigma_2$:

$$\int d^2x^\pm \, g(x^\pm)^2 \, \langle \phi^2(x^+, x^-, \vec{y}=0)\rangle_\psi \geq -\min_D \mathcal{Q}_{\phi\phi}[D]. \tag{56}$$

We start with the "partial Fourier transform" of the vacuum correlator

$$\langle \phi(k_+, k_-, \vec{y}=0)\phi(k_+', k_-', \vec{y}=0)\rangle_\Omega = (2\pi)^2 \delta^2(k+k')G_{\phi\phi}(k'), \tag{57}$$

with

$$G_{\phi\phi}(k) = \frac{V_{n-3}}{2(2\pi)^{n-3}}(4k_+k_- - m^2)^{\frac{n-4}{2}}\Theta\left(4k_+k_- - m^2\right)\Theta(k_+). \tag{58}$$

Writing

$$\mathcal{Q}_{\phi\phi}[D] = \int \frac{d^2k_\pm}{(2\pi)^2}|\tilde{g}(k_\pm)|^2 h_D(k_\pm) \tag{59}$$

then by similar techniques to the previous section, for the boosted domains, $D_\eta$ we find

$$h_{D_\eta}(k_\pm) = \frac{4}{(4\pi)^{n/2}}\Theta(k_\eta - m)k_\eta^{n-2}\int_{m^2/k_\eta^2}^1 dw \left(w - \frac{m^2}{k_\eta^2}\right)^{\frac{n-4}{2}}\log\left(\frac{1+\sqrt{1-w}}{1-\sqrt{1-w}}\right), \tag{60}$$

where we changed variables to $w = 4\zeta_+\zeta_-/k_\eta^2$. When the field is massless, the dependence on $k_\eta$ is completely power-law and the integral can be evaluated when exactly for $n \geq 3$ (when $n = 2$ the integral diverges leaving the bound trivial). The end result is

$$\boxed{\int_{\Sigma_2} d^2x^\pm \, g(x^\pm)^2 \langle \phi^2(x^\pm, \vec{y}=0)\rangle_\psi \bigg|_{m^2=0} \geq -\min_{\eta\in\mathbb{R}} c_{\phi\phi}^{(n)} \int \frac{d^2k_\pm}{(2\pi)^2}|\tilde{g}(k_\pm)|^2 \, k_\eta^{n-2}\Theta(k_\eta),} \tag{61}$$

with

$$c_{\phi\phi}^{(n)} = \frac{4}{(4\pi)^{\frac{n-1}{2}}}\left(\frac{n-4}{n-2}\right)\frac{\Gamma\left(\frac{n-4}{2}\right)}{\Gamma\left(\frac{n-2}{2}\right)\Gamma\left(\frac{n-1}{2}\right)}, \tag{62}$$

and $k_\eta = e^\eta k_+ + e^{-\eta}k_-$. To our knowledge, neither this bound (or its massive counterpart, (60)) have appeared explicitly in the literature.

## 3.4 Smeared higher-spin currents

As a final example, we use (30) to derive a similar double-smeared bound on null higher-spin currents. To be specific, we will focus on the *massless* scalar (again in $n$ dimensions). As is well-known, there are a tower of even-spin conserved currents with null-components given by

$$\mathcal{J}_{--\ldots-} =: \partial_-^{\mathfrak{s}/2}\phi \, \partial_-^{\mathfrak{s}/2}\phi : \qquad \mathfrak{s} \text{ even}, \tag{63}$$

up to total derivative. Given the above recipe, bounding $\mathcal{J}_{--\ldots-}$ only amounts a modification in the kernel $G$:

$$G_{\mathcal{J}_{--\ldots-}}(k) = \frac{V_{n-3}}{2(2\pi)^{n-3}} \, k_-^{\mathfrak{s}}(4k_+k_-)^{\frac{n-4}{2}}\Theta(k_+)\Theta(k_-). \tag{64}$$

By similar mathematics as above, the associated $h_{D_\eta}(k)$ can be evaluated as

$$h_{D_\eta}(k) = \frac{4}{(4\pi)^{n/2}}\int dq_- du \, q_-^{\mathfrak{s}-1}u^{\frac{n-4}{2}}\Theta(\zeta_-)\Theta(u)\Theta\left(k_\eta - e^{-\eta}\zeta_- - e^\eta\frac{u}{4\zeta_-}\right)$$

$$= \frac{e^{\mathfrak{s}\eta}}{2^{\mathfrak{s}-4}(4\pi)^{n/2}\mathfrak{s}}\left(\sum_{\ell \text{ odd}}^{\mathfrak{s}-1}\binom{\mathfrak{s}}{\ell}\frac{\Gamma\left(\frac{\ell+2}{2}\right)}{\Gamma\left(\frac{\ell+n}{2}\right)}\right)k_\eta^{\mathfrak{s}+n-2}\Theta(k_\eta), \tag{65}$$

leading to a bound of the following form

$$\int_{\Sigma_2} d^2x^\pm \, g(x^\pm)^2 \, \langle \mathcal{J}_{--\ldots-}(x^\pm, \vec{y}_\perp = 0) \rangle_\psi \geq -\min_{\eta \in \mathbb{R}} c_{\mathfrak{s}}^{(n)} \int \frac{d^2k_\pm}{(2\pi)^2} |\tilde{g}(k)|^2 \, e^{\mathfrak{s}\eta} \, k_\eta^{\mathfrak{s}+n-2} \Theta(k_\eta), \qquad (66)$$

with

$$c_{\mathfrak{s}}^{(n)} = \frac{1}{2^{\mathfrak{s}-4}(4\pi)^{n/2}\,\mathfrak{s}} \sum_{\substack{k \text{ odd}}}^{\mathfrak{s}-1} \binom{\mathfrak{s}}{k} \frac{\Gamma\left(\frac{k+2}{2}\right)}{\Gamma\left(\frac{k+n}{2}\right)}, \qquad (67)$$

and again, $k_\eta = e^\eta k_+ + e^{-\eta} k_-$.

**Higher Spin ANEC**

We can put (66) in a similar form to that of the DSNEC by again defining the smearing lengths explictly in our smearing function,

$$g(x^+, x^-) = \frac{1}{\sqrt{\delta^+ \delta^-}} \mathcal{F}(x^+/\delta^+, x^-/\delta^-), \qquad (68)$$

for some dimensionless smooth function $\mathcal{F}(s^+, s^-)$ dropping off quickly for $|s^\pm| \gg 1$. Rescaling our boost parameter $e^{\tilde{\eta}} := \sqrt{\frac{\delta^-}{\delta^+}} e^\eta$ and integration variable $\rho_\pm = \delta^\pm k_\pm$ the bound takes the schematic form

$$\int \frac{d^2x^\pm}{\delta^+\delta^-} \mathcal{F}(x^+/\delta^+, x^-/\delta^-)^2 \langle \mathcal{J}_{--\ldots-}(x^\pm, \vec{y}_\perp = 0) \rangle_\psi \geq -\frac{\mathcal{N}_{\mathfrak{s},n}}{(\delta^+)^{\frac{n-2}{2}}(\delta^-)^{\frac{n-2}{2}+s}}, \qquad (69)$$

where $\mathcal{N}_{\mathfrak{s},n}$ is an O(1) dimensionless factor depending on the details of the smearing function:

$$\mathcal{N}_{\mathfrak{s},n} = \min_{\tilde{\eta} \in \mathbb{R}} c_{\mathfrak{s}}^{(n)} \int \frac{d^2\rho_\pm}{(2\pi)^2} \, e^{s\tilde{\eta}} |\tilde{\mathcal{F}}(\rho_\pm)|^2 \rho_{\tilde{\eta}}^{s+n-2} \Theta(\rho_{\tilde{\eta}}). \qquad (70)$$

Now once again we can let $\mathcal{F}(s^+, s^-) = f_{\sqrt{G}}(s^+) f_{\sqrt{G}}(s^-)$ factorize where $f_{\sqrt{G}}$ is the square-root of the normalized Gaussian with unit variance, (8). We then multiply both sides of the bound by $\delta^-$ and take the limit $\delta^+ \to 0$ while holding $\alpha \equiv \delta^+ \delta^-$ fixed. In this limit we recover the "higher spin ANEC" proposed by [3]

$$\int dx^- \langle \mathcal{J}_{--\ldots-}(x^+ = 0, x^-, \vec{y}_\perp = 0) \rangle_\psi \geq -\lim_{\delta^+ \to 0} \frac{\sqrt{2\pi} \mathcal{N}_{\mathfrak{s},n}}{\alpha^{\frac{n-2}{2}+s-1}} (\delta^+)^{\mathfrak{s}-1} = 0. \qquad (71)$$

# 4 Worldvolume QNEI

Having explained the method for deriving the DSNEC (among other bounds) as difference inequalities in Minkowski space, we will show in this section how this bound is implied by an existing absolute QEI [4] averaged over a spacetime worldvolume. This QEI is valid for general curved spacetimes, however here we focus on Minkowski space. First, we will describe the QEI in question. Then, we use it to obtain a familiar timelike averaged bound first derived by Fewster and Roman [17] in 4 dimensions as a pedagogical example. We will then proceed to apply the QEI to the DSNE, confirming the bounds described in Sec. 3.2.

## 4.1 A general quantum null energy inequality

We start by stating the general form of the QEI of Ref. [4]:

$$\int_{\Sigma} d\text{vol}(x)g^2(x)\left[\!\left[\mathcal{D}\otimes\mathcal{D}(W_\Psi - H_{(k)})\right]\!\right] \geq$$
$$-2\int_{\mathcal{D}} \frac{d^n\xi}{(2\pi)^n}\left[|h_\kappa|^{1/4}g_\kappa \otimes |h_\kappa|^{1/4}g_\kappa \vartheta_\kappa^*(\mathcal{D}\otimes\mathcal{D}\tilde{H}_{(k)})\right]^\wedge(-\xi,\xi), \qquad (72)$$

where $\mathcal{D}$ is a partial differential operator of order at most one with smooth real-valued coefficients. For convenience, we have introduced a notation $[\cdot]^\wedge(\xi_1,\xi_2)$ for a bi-Fourier transform in two arguments $x$ and $x'$, i.e.

$$[B]^\wedge(\xi_1,\xi_2) := \int d^n x\, d^n x'\, e^{i\xi_1 x + \xi_2 x'} B(x,x'). \qquad (73)$$

The Hadamard bi-distribution $H$, expressed as an infinite series in even dimensions, is given by [30]

$$H(x,x') = \frac{\Gamma\left(\frac{n-2}{2}\right)}{4\pi^{n/2}}\left\{\frac{U(x,x')}{\sigma_+(x,x')^{n/2-1}} + V(x,x')\ln\left[\frac{\sigma_+(x,x')}{\ell^2}\right] + W(x,x')\right\}, \qquad (74)$$

where $\ell$ is an arbitrary length scale, and for odd dimensions

$$H(x,x') = \frac{\Gamma\left(\frac{n-2}{2}\right)}{4\pi^{n/2}}\left\{\frac{U(x,x')}{\sigma_+(x,x')^{n/2-1}} + W(x,x')\right\}. \qquad (75)$$

The bi-distributions $U(x,x')$ and $V(x,x')$ are regular in the coincidence limit and can be expressed as power-series in $\sigma$

$$U(x,x') = \sum_{\ell=0}^{\infty} U_\ell(x,x')\sigma(x,x')^\ell, \qquad V(x,x') = \sum_{\ell=0}^{\infty} V_\ell(x,x')\sigma(x,x')^\ell, \qquad (76)$$

with symmetric coefficients calculated uniquely by requiring that $H$ obeys the field equation (9) at each order with the appropriate boundary conditions [30]. In contrast,

$$W(x,x') = \sum_{\ell=0}^{\infty} W_\ell(x,x')\sigma(x,x')^\ell \qquad (77)$$

are not uniquely specified as $W_0(x)$ is undetermined. This coefficient depends on the state of the quantum field and once it is fixed the $W_\ell$'s can also be determined using the recursion relations derived from the field equation.

We will specify the order of the Hadamard series using the following convention: by $H_k$ we denote the term of order[6] $\sigma^k$ while by $H_{(k)}$ all the terms up to order $k$. In Ref. [4], it was required that $k = \max\{n+3, 5\}$ for the QEI of Eq. (72). However, in Ref. [32] it was shown that only $k = 2$ are needed for a first order differential operator.

We define

$$\tilde{H}(x,x') = \frac{1}{2}[H(x,x') + H(x',x) + iE(x,x')], \qquad (78)$$

where $E(x,x')$ is the antisymmetric part of the two-point function.

---

[6]By convention terms of the form $\log\sigma$ are order zero.

The function $\sigma$ is the squared invariant length of the geodesic between $x$ and $x'$, negative for timelike separation. In flat space

$$\sigma(x, x') = -\eta_{\mu\nu}(x - x')^{\mu}(x - x')^{\nu}, \tag{79}$$

where $\eta_{\mu\nu}$ is the Minkowski metric. By $F(\sigma_+)$, for some distribution $F$, we mean the distributional limit

$$F(\sigma_+) = \lim_{\epsilon \to 0^+} F(\sigma_\epsilon), \tag{80}$$

where

$$\sigma_\epsilon(x, x') = \sigma(x, x') + 2i\epsilon(t(x) - t(x')) + \epsilon^2. \tag{81}$$

Following Ref. [4] and [33] we define a *small sampling domain*. A small sampling domain $\Sigma$ is defined to be an open subset of $(\mathcal{M}, g)$ that (i) is contained in a globally hyperbolic convex normal neighbourhood of $M$, (ii) may be covered by a single *hyperbolic coordinate chart* $\{x^{\mu}\}$, which requires that $\partial/\partial x^0$ is future pointing and timelike and that there exists a constant $c > 0$ such that

$$c|u_0| \geq \sqrt{\sum_{j=0}^{3} u_j^2} \tag{82}$$

holds for the components of every causal covector, $u$, at each point of $\Sigma$. That statement means that the coordinate speed of light is bounded. Now we may express the hyperbolic chart $\{x^{\mu}\}$ by map $\kappa : \Sigma \to \mathbb{R}^n$, where, $\kappa(p) = (x^0(p), x^1(p), \ldots, x^{n-1}(p))$. Any function $g$ on $\Sigma$ determines a function $g_\kappa = g \circ \kappa^{-1}$ on $\Sigma_\kappa = \kappa(\Sigma)$. In particular, the inclusion map $\iota : \Sigma \to \mathcal{M}$ induces a smooth map $\iota_\kappa : \Sigma_\kappa \to \mathcal{M}$. We have $\vartheta : \Sigma \times \Sigma \to \mathcal{M} \times \mathcal{M}$ the map $\vartheta(x, x') = (\iota \otimes \iota)(x, x')$. Here $h = \iota^* g$ is a Lorentzian metric on $\Sigma$ and $h_\kappa$ is the determinant of the matrix $\kappa^* h$. Then the bundle $\mathcal{N}^+$ of non-zero future pointing null covectors on $(M, g)$ pulls back under $\iota_\kappa$ so that

$$\iota_\kappa^* \mathcal{N}^+ \subset \Sigma_\kappa \times D, \tag{83}$$

where $D \subset \mathbb{R}^n$ is the set of all $u_a$ that satisfy Eq. (82). As in Minkowski space there is some freedom in choosing $D$. One example of appropriate $D$ is $D_0$ which is the set of all $u_a$ with $u_0 > 0$ so it is a proper subset of the upper half space $\mathbb{R}^+ \times \mathbb{R}^{n-1}$.

To conclude the introduction of this general QEI we should note that it is not covariant in full generality because it depends on the coordinates used and the choice of tetrad near $\Sigma$. Ref. [4] following methods of [34] showed that covariance can be restored by picking the right set of coordinates. In particular with a choice of Fermi normal coordinates the bound was shown to be locally covariant.

Now we state a specific example of the QEI of Eq. (72) where the differential operator $\mathcal{D} = \ell^{\mu}\nabla_{\mu}$, where $\ell^{\mu}$ is a future pointing null vector, thus the quantity bounded is the null energy. The bound has the form

$$\int_{\mathcal{M}} d\mathrm{vol}g^2(x) \langle T_{\mu\nu}^{\mathrm{ren}} \ell^{\mu}\ell^{\nu} \rangle_\psi \geq$$
$$-2 \int_D \frac{d^n\xi}{(2\pi)^n} \left[ |h_\kappa|^{1/4} g_\kappa \otimes |h_\kappa|^{1/4} g_\kappa \left( \left( \ell^{\mu}\nabla_{\mu} \otimes \ell^{\nu'}\nabla_{\nu'}\tilde{H}_{(2)} \right)_\kappa \right]^{\wedge} (-\xi, \xi)$$
$$+ \int_{\mathcal{M}} d\mathrm{vol}g(x)^2 C_{\mu\nu}\ell^{\mu}\ell^{\nu}. \tag{84}$$

From this inequality one can calculate the bound for general curved spacetimes in a perturbative way. By perturbative here we mean up to a certain order in the curvature components and their derivatives.

A similar calculation was performed in [32] for the energy density in $n = 4$ dimensions and it included the computation of the Hadamard coefficients in $\tilde{H}_{(2)}$. If additionally one requires that the curvature components and their derivatives are finite (but not necessarily small) the bound can be written as a sum of integrals of the smearing function with constant coefficients. For example, one such additional term that appears in the energy density bound is (schematically)

$$\int dt f(t)^2 \langle T_{00}^{\text{ren}} \rangle_\psi \gtrsim \dots - R_{\max} \int f'(t)^2 dt + \dots, \quad \text{where} \quad |R_{\mu\nu}| \leq R_{\max}, \tag{85}$$

in a Fermi-Walker coordinate system [32]. Then the derivation of DSNEC for curved spacetimes can follow the procedure described in the following subsections for Minkowski spacetime.

**Massless fields**

Let us first discuss the bound for massless fields in Minkowski space for which Eq. (84) becomes

$$\int_{\mathcal{M}} d\text{vol} g(x)^2 \langle T_{\mu\nu}^{\text{ren}} \ell^\mu \ell^\nu \rangle_\psi \geq -2 \int_D \frac{d^n\xi}{(2\pi)^n} \left[ (g\ell^\mu \nabla_\mu \otimes g\ell^{\nu'} \nabla_{\nu'}) \tilde{H}_{-n/2+1} \right]^{\wedge} (-\xi, \xi), \tag{86}$$

where only the most singular term of (74)-(75) is relevant:

$$\tilde{H}_{-n/2+1}(x, x') = H_{-n/2+1}(x, x') = \frac{\Gamma\left(\frac{n-2}{2}\right)}{4\pi^{n/2}} \frac{1}{\sigma_+(x, x')^{n/2-1}}. \tag{87}$$

Since we are in Minkowski space the timelike curve can be parametrized by $t$. Then we can define $\Delta t = t - t'$ and $\Delta \vec{x} = \vec{x} - \vec{x}'$

$$H_{-n/2+1}(x, x') = \frac{(-1)^{n/2-1} \Gamma\left(\frac{n-2}{2}\right)}{4\pi^{n/2}((\Delta t - i\epsilon)^2 - |\Delta \vec{x}|^2)^{n/2-1}}. \tag{88}$$

To proceed we pick the direction $(-)$ for the null vector $\ell^\mu$ defined by $x^- = t - x$ while $x^+ = t + x$. Then

$$\ell^\mu \partial_\mu = \partial_- = \frac{1}{2}(\partial_t - \partial_x), \tag{89}$$

and

$$H_{-n/2+1}(x, x') = \frac{(-1)^{n/2-1} \Gamma\left(\frac{n-2}{2}\right)}{4\pi^{n/2}(\Delta x^- \Delta x^+ - 2i\epsilon \Delta t - \Delta y^2)^{n/2-1}}, \tag{90}$$

where we remind the reader that $y$ denotes the transverse spatial dimensions. Applying the derivatives gives

$$(\ell^\mu \partial_\mu)(\ell^{\nu'} \partial_{\nu'}) \tilde{H}_{-n/2+1}(x, x') = \frac{(-1)^{n/2} \Gamma\left(\frac{n+2}{2}\right)(\Delta x^+)^2}{4\pi^{n/2}(\Delta x^- \Delta x^+ - 2i\epsilon \Delta t - \Delta y^2)^{n/2+1}}. \tag{91}$$

So we have

$$\int_{\mathcal{M}} d\text{vol} g(x)^2 \langle T_{--}^{\text{ren}} \rangle_\psi \geq$$
$$-\frac{(-1)^{n/2} \Gamma\left(\frac{n+2}{2}\right)}{2\pi^{n/2}} \int_D \frac{d^n\xi}{(2\pi)^n} \left[ g(x)g(x') \frac{(\Delta x^+)^2}{(\Delta x^- \Delta x^+ - 2i\epsilon \Delta t - \Delta y^2)^{n/2+1}} \right]^{\wedge} (-\xi, \xi). \tag{92}$$

**Massive fields**

For massive fields there are additional singular terms in the Hadamard parametrix that are relevant for the renormalization of the stress tensor. The details of these terms are dimension dependent and are fixed requiring the expansion (76) to satisfy the massive field equation at each order. Likewise the number of terms relevant for renormalizing the stress tensor is dimension dependent. In particular, following the standard Hadamard renormalization prescription of subtracting only the singular terms (what one might regard as a *minimal subtraction scheme*) only a finite number of coefficients $U_\ell$ and $V_\ell$ are relevant. We emphasize that in Minkowski space this is, in principle, a different scheme than normal ordering: since the Minkowski two-point function is typically a transcendental function of the mass, subtraction by the vacuum expectation value is a subtraction in all orders of $m^2$. Since these two schemes differ only by terms vanishing in the coincident limit they yield the same local operator, $T_{--}^{\mathrm{ren}}$, however for the derivation of a lower bound we will find different results. To see this, in section 4.3 we will construct the corresponding lower bound implied by (84) for the 4d massive scalar and discuss it in comparison to the Minkowski difference inequality (50).

## 4.2 Timelike smearing

As a brief check on the content of (92), let us reproduce the time-like null-energy bound in 4d first derived in [17]; this will also provide a blue-print calculation for the double smeared quantities to follow. To implement a worldline smearing we will take

$$g(t, \vec{x})^2 = g_0(t)^2 \delta^3(\vec{x}). \tag{93}$$

To take the "square-root" of the delta function, we will regard it is as the limit of a sharply-peaked Gaussian, i.e.

$$g(t, \vec{x}) = \lim_{\sigma \to 0} g_0(t) \frac{1}{\sigma^{3/2}(2\pi)^{3/4}} e^{-\frac{\vec{x}^2}{4\sigma^2}}, \tag{94}$$

where its Fourier transform is given by

$$\tilde{g}(\omega, \vec{k}) = \lim_{\sigma \to 0} \tilde{g}_0(\omega) \sigma^{3/2} \pi^{3/4} 2^{9/4} e^{-\sigma^2 \vec{k}^2}. \tag{95}$$

From Eq. (92) for $n = 4$, The bound on the time-like smeared null-energy then is

$$\int dt\, g_0(t)^2 \langle T_{--}^{\mathrm{ren}}(t) \rangle_\psi \geq$$

$$-\lim_{\sigma \to 0} \frac{\sigma^3 2^{9/2}}{\pi^{1/2}} \int_D \frac{d^4\xi}{(2\pi)^4} \int \frac{d\omega}{(2\pi)} \frac{d\omega'}{(2\pi)} \int \frac{d^3\vec{k}}{(2\pi)^3} \frac{d^3\vec{k}'}{(2\pi)^3} \int d^4x\, d^4x'\, e^{-\sigma^2 \vec{k}^2 - \sigma^2 \vec{k}'^2}$$

$$e^{i\omega t + i\omega' t'} e^{i\vec{k}\vec{x} + i\vec{k}'\vec{x}'} e^{-i\xi \Delta x} \tilde{g}_0(\omega) \tilde{g}_0(\omega') \frac{(\Delta x^+)^2}{(\Delta x^- \Delta x^+ - 2i\epsilon \Delta t - \Delta y^2)^3}. \tag{96}$$

We will shift the $x$ integral to $s \equiv \Delta x = x - x'$. The $x'$ integral yields the delta functions $(2\pi)^4 \delta(\omega + \omega') \delta^3(\vec{k} + \vec{k}')$, which we then collapse and perform the Gaussian integration over $\vec{k}$:

$$\int dt\, g_0(t)^2 \langle T_{--}^{\mathrm{ren}}(t) \rangle_\psi \geq -\lim_{\sigma \to 0} \frac{1}{\pi^2} \int_D \frac{d^4\xi}{(2\pi)^4} \int \frac{d\omega}{(2\pi)} |\tilde{g}_0(\omega)|^2$$

$$\times \int d^4s\, e^{i\omega s_0 - i\xi s} e^{-\frac{\vec{s}^2}{8\sigma^2}} \frac{(s^+)^2}{(s^- s^+ - 2i\epsilon s_0 - s_y^2)^3}. \tag{97}$$

We now make a specific choice of smooth sampling domain, namely $D_0 = \{\xi_0 \geq 0\}$. The integration over $\vec{\xi}$ then introduces $(2\pi)^3 \delta^3(\vec{s})$ which, upon collapsing, leaves the $\sigma \to 0$ limit safe:

$$\int dt\, g_0(t)^2 \langle T_{--}^{\text{ren}}(t) \rangle_\psi \geq -\frac{1}{\pi^2} \int_0^\infty \frac{d\xi_0}{2\pi} \int_{-\infty}^\infty \frac{d\omega}{2\pi} |\tilde{g}_0(\omega)|^2 \int ds_0 \frac{1}{(s_0 - i\epsilon)^4} e^{i(\omega - \xi_0)s_0}. \quad (98)$$

Using the general result

$$\lim_{\epsilon \to 0} \int_{-\infty}^\infty ds\, \frac{e^{i\omega s}}{(s - i\epsilon)^p} = \frac{2\pi e^{ip\pi/2}}{\Gamma(p)}\, \omega^{p-1}\, \Theta(\omega), \qquad p \in \mathbb{R}, \quad (99)$$

and writing $\zeta = \omega - \xi_0$ we have

$$\int dt\, g_0(t)^2 \langle T_{--}^{\text{ren}}(t) \rangle_\psi \geq -\frac{1}{3\pi} \int_0^\infty \frac{d\omega}{2\pi} |\tilde{g}_0(\omega)|^2 \int_0^\omega \frac{d\zeta}{2\pi} \zeta^3 = -\frac{1}{24\pi^2} \int_0^\infty \frac{d\omega}{2\pi} |\tilde{g}_0(\omega)|^2 \omega^4. \quad (100)$$

Using $\widetilde{g_0'}(\omega) = i\omega \tilde{g}_0(\omega)$ and Parseval's theorem

$$\int_0^\infty \frac{d\omega}{2\pi} |\tilde{g}_0(\omega)|^2 \omega^4 = \int_0^\infty \frac{d\omega}{2\pi} |\widetilde{g_0''}(\omega)|^2 = \frac{1}{2} \int_{-\infty}^\infty \frac{d\omega}{2\pi} |\widetilde{g_0''}(\omega)|^2 = \frac{1}{4} \int_{-\infty}^\infty dt\, g_0''(t)^2, \quad (101)$$

we arrive at a nice representation in position space, matching the result of [4]

$$\boxed{\int dt\, g_0(t)^2 \langle T_{--}^{\text{ren}}(t) \rangle_\psi \geq -\frac{(\ell_- \cdot \partial_t)^2}{12\pi^2} \int_{-\infty}^\infty dt\, g_0''(t)^2,} \quad (102)$$

where $\ell_- \cdot \partial_t = \eta_{\mu\nu}(\partial_-)^\mu(\partial_t)^\nu = \frac{1}{2}$.

## 4.3 Double null smearing

Now we want to apply the worldvolume QEI, (92), to the main object of interest, the DSNE, in $n$ spacetime dimensions. To do so we write the smearing function as

$$g(x^+, x^-, \vec{y})^2 = g(x^+, x^-)^2 \delta^{n-2}(\vec{y}), \quad (103)$$

and so

$$g(x^+, x^-, \vec{y}) = \lim_{\sigma \to 0} g(x^+, x^-) \frac{1}{\sigma^{(n-2)/2}(2\pi)^{(n-2)/4}} e^{-|\vec{y}|^2/4\sigma^2}. \quad (104)$$

Then the bound of (92) can be written as

$$\int d^2x^\pm g(x^\pm)^2 \langle T_{--}^{\text{ren}} \rangle_\psi \geq -\lim_{\sigma \to 0} \frac{\Gamma\left(\frac{n+2}{2}\right) 2^{3n/2} \sigma^{n-2}}{(-1)^{n/2}\pi} \int_D \frac{d^n\xi}{(2\pi)^n}$$
$$\int \frac{d^2k_\pm d^2k_\pm'}{(2\pi)^4} \tilde{g}(k_+, k_-)\tilde{g}(k_+', k_-') \int \frac{d^{n-2}\vec{k}_y d^{n-2}\vec{k}_y'}{(2\pi)^{2(n-2)}} e^{-\sigma^2|\vec{k}_y|^2 - \sigma^2|\vec{k}_y'|^2}$$
$$\int d^nx\, d^nx'\, e^{ikx + ik'x'} e^{-i\xi(x-x')} \frac{(\Delta x^+)^2}{(\Delta x^- \Delta x^+ - 2i\epsilon\Delta t - \Delta y^2)^{n/2+1}}. \quad (105)$$

A calculation following a wholly similar logic as section 4.2 (we refer the reader interested in following the details to look there) leads to

$$\int d^2x^\pm g(x^\pm)^2 \langle T_{--}^{\text{ren}} \rangle_\psi \geq -\frac{2\Gamma\left(\frac{n+2}{2}\right)}{(-1)^{n/2}\pi^{n/2}} \int_D \frac{d^2\xi_\pm}{(2\pi)^2} \int \frac{d^2k_\pm}{(2\pi)^2} |\tilde{g}(k_+, k_-)|^2$$
$$\times \int d^2s^\pm e^{i(k-\xi)_\pm s^\pm} \frac{1}{(s^+ - i\epsilon)^{n/2-1}(s^- - i\epsilon)^{n/2+1}}. \quad (106)$$

Utilizing (99) we arrive at

$$
\int d^2x^\pm g(x^\pm)^2 \langle T_{--}^{\mathrm{ren}}\rangle_\psi \geq -\frac{8}{\pi^{n/2-2}\Gamma\left(\frac{n-2}{2}\right)}\int_D \frac{d^2\xi_\pm}{(2\pi)^2}\int \frac{d^2k_\pm}{(2\pi)^2}|\tilde{g}(k_+,k_-)|^2
$$
$$
\times (k_+ - \xi_+)^{n/2-2}\Theta(k_+ - \xi_+)(k_- - \xi_-)^{n/2}\Theta(k_- - \xi_-).
$$
(107)

Up to now we have been fairly agnostic about the domain, $D$, beyond that it satisfies the criteria of a small sampling domain. The freedom to choose this domain is very much analogous to the choice of domain we encountered in the difference inequalities in section 3. In principle this freedom of a small sampling domain is a parameter that can be optimized, however, much like in section 3 we restrict our focus to boosted domains of the form $D_\eta := \{\xi_\eta = e^\eta \xi_+ + e^{-\eta}\xi_- \geq 0\}$ and optimizing over $\eta \in \mathbb{R}$.

Defining variables $\zeta_\pm = k_\pm - \xi_\pm$ (with the constraint $k_\eta - \zeta_\eta \geq 0$ due to the domain $D_\eta$)

$$
\int d^2x^\pm g(x^\pm)^2 \langle T_{--}^{\mathrm{ren}}\rangle_\psi \geq -\frac{2}{\pi^{n/2}\Gamma\left(\frac{n-2}{2}\right)}\int \frac{d^2k_\pm}{(2\pi)^2}|\tilde{g}(k_+,k_-)|^2
$$
$$
\int_0^{e^\eta k_\eta} d\zeta_- \int_0^{e^{-\eta}k_\eta - e^{-2\eta}\zeta_-} d\zeta_+ (\zeta_+)^{n/2-2}(\zeta_-)^{n/2}\Theta(k_\eta).
$$
(108)

The $\zeta_\pm$ integrals then give the final expression

$$
\boxed{\int d^2x^\pm g(x^+,x^-)^2 \langle T_{--}^{\mathrm{ren}}\rangle_\psi \geq -\frac{e^{2\eta}}{(4\pi)^{\frac{n-1}{2}}\Gamma\left(\frac{n+1}{2}\right)}\int \frac{d^2k_\pm}{(2\pi)^2}|\tilde{g}(k_+,k_-)|^2 k_\eta^n \Theta(k_\eta),}
$$
(109)

with $k_\eta = e^\eta k_+ + e^{-\eta}k_-$. This is the same expression as the one derived in Sec. 3.2 (in the massless limit).

**Including a mass**

We can compute the mass corrections, in 4d, to the massless bound, (109), by using the expansion of the Hadamard parametrix. The relevant terms for the 4d massive scalar in Minkowski spacetime are given by

$$
H_{(1)}(x,x') = H_{-1}(x,x') + H_0(x,x') + H_1(x,x'),
$$
(110)

where

$$
\tilde{H}_{-1}(x,x') = H_{-1}(x,x') = -\frac{1}{4\pi^2}\frac{1}{(\Delta x^- \Delta x^+ - 2i\epsilon\Delta t - \Delta y^2)},
$$
$$
\tilde{H}_0(x,x') = H_0(x,x') = -\frac{v_0}{4\pi^2}\ln\left((\Delta x^- \Delta x^+ - 2i\epsilon\Delta t - \Delta y^2)/\ell^2\right),
$$
$$
\tilde{H}_1(x,x') = H_1(x,x') = -\frac{v_1}{4\pi^2}(\Delta x^- \Delta x^+ - \Delta y^2)\ln\left((\Delta x^- \Delta x^+ - 2i\epsilon\Delta t - \Delta y^2)/\ell^2\right),
$$
(111)

with coefficients[7] [30]

$$
v_0 = -\frac{1}{4}m^2, \qquad v_1 = \frac{1}{32}m^4.
$$
(112)

---

[7]These coefficients differ slightly from [30] due to a difference in definition in $\sigma(x,x')$.

Higher order terms vanish in the coincidence limit. Applying the derivatives gives

$$(\partial_-)(\partial'_-)\tilde{H}_{(1)}(x,x') = \frac{(\Delta x^+)^2}{2\pi^2(\Delta x^-\Delta x^+ - 2i\epsilon\Delta t - \Delta y^2)^3}$$
$$+ \frac{m^2(\Delta x^+)^2}{16\pi^2(\Delta x^-\Delta x^+ - 2i\epsilon\Delta t - \Delta y^2)^2} + \frac{m^4(\Delta x^+)^2}{128\pi^2(\Delta x^-\Delta x^+ - 2i\epsilon\Delta t - \Delta y^2)}. \quad (113)$$

Tracking the terms through the calculation of the previous section we arrive, intermediately, at

$$\int d^2x^{\pm}g(x^+,x^-)^2\langle T^{\mathrm{ren}}_{--}\rangle_\psi \geq -\frac{4}{\pi^2}\int_{D_\eta}\frac{d^2\xi_\pm}{(2\pi)^2}\int\frac{d^2k_\pm}{(2\pi)^2}|\tilde{g}(k_+,k_-)|^2\int d^2s^{\pm}e^{i(k-\xi)_\pm s^{\pm}}$$
$$\times\left\{\frac{1}{(s^+ - i\epsilon)(s^- - i\epsilon)^3} + \frac{m^2}{8}\frac{1}{(s^- - i\epsilon)^2} + \frac{m^4}{64}\frac{s^+}{(s^- - i\epsilon)}\right\}$$
$$\geq -8\int\frac{d^2\zeta_\pm}{(2\pi)^2}\int\frac{d^2k_\pm}{(2\pi)^2}|\tilde{g}(k_+,k_-)|^2\,\Theta(k_\eta - \zeta_\eta)$$
$$\times\left\{\zeta_-^2\,\Theta(\zeta_-)\Theta(\zeta_+) - \frac{m^2}{4}\delta(\zeta_+)\zeta_-\Theta(\zeta_-) + \frac{m^4}{32}\delta'(\zeta_+)\Theta(\zeta_-)\right\}, \quad (114)$$

where in the second line we changed variables to $\zeta_\pm = k_\pm - \xi_\pm$ and incorporated the boosted domain, $D_\eta$ into an appropriate theta function. From here the $\zeta_\pm$ integrals are simple to do (noting that $\frac{\partial}{\partial\zeta_+}\Theta(k_\eta - \zeta_\eta) = -e^\eta\delta(k_\eta - \zeta_\eta)$)

$$\int d^2x^{\pm}g(x^+,x^-)^2\langle T^{\mathrm{ren}}_{--}\rangle_\psi \geq -\frac{e^{2\eta}}{6\pi^2}\int\frac{d^2k_\pm}{(2\pi)^2}|\tilde{g}(k_+,k_-)|^2\left(k_\eta^4 - \frac{3}{2}m^2k_\eta^2 + \frac{3}{8}m^4\right)\Theta(k_\eta). \quad (115)$$

We note that this is a different lower bound for massive fields than what we found by direct construction in Minkowski space, (50). As discussed at the end of section 4.1, this is somewhat expected since Hadamard renormalization only subtracts a finite number of singular terms while normal ordering subtracts an expectation value containing all orders in a mass expansion. To check this intuition we can compare (115) to a perturbative expansion of (50)

$$\int d^2x^{\pm}g(x^+,x^-)^2\langle T^{\mathrm{ren}}_{--}\rangle_\psi \geq -\frac{e^{2\eta}}{6\pi^2}\int\frac{d^2k_\pm}{(2\pi)^2}|\tilde{g}(k_+,k_-)|^2\left\{\left(k_\eta^4 - \frac{3}{2}k_\eta^2 m^2 + \frac{3}{8}m^4\right)\Theta(k_\eta)\right.$$
$$\left. + k_\eta^4\left(-m\delta(k_\eta) + \frac{m^2}{2}\delta'(k_\eta) - \frac{m^3}{6}\delta''(k_\eta) + \frac{m^4}{24}\delta'''(k_\eta)\right) + \ldots\right\}$$
$$\geq -\frac{e^{2\eta}}{6\pi^2}\int\frac{d^2k_\pm}{(2\pi)^2}|\tilde{g}(k_+,k_-)|^2\left\{\left(k_\eta^4 - \frac{3}{2}k_\eta^2 m^2 + \frac{3}{8}m^4\right)\Theta(k_\eta) + \ldots\right\}, \quad (116)$$

where the second line, coming from expanding $\Theta(k_\eta - m)$, vanishes up to order $m^4$. This expansion matches (115) up to the "..." indicating higher order mass terms. Note that while the coefficients of the mass terms can come with either sign, the general expectation is that full difference inequality (50) is a qualitatively *stronger* lower bound than (115). Indeed it was shown in [1] that for Gaussian smearing functions of smearing lengths $\delta^{\pm}$, that at large mass the integral in (50) can be evaluated at saddle-point and is exponentially suppressed in $m^2\delta^+\delta^-$ making the bound very tight. This is contrast to (115), which gets weaker as the mass increases.

To illustrate that fact we write the bound as a function of $\gamma := (\delta^+\delta^- m^2)^{1/2}$

$$\int d^2x^{\pm}g(x^{\pm})^2\langle T^{\mathrm{ren}}_{--}\rangle_\psi \geq -\frac{c^{(4)}_{T_{--}}}{\delta^+(\delta^-)^3}\tilde{Q}(\gamma), \quad (117)$$

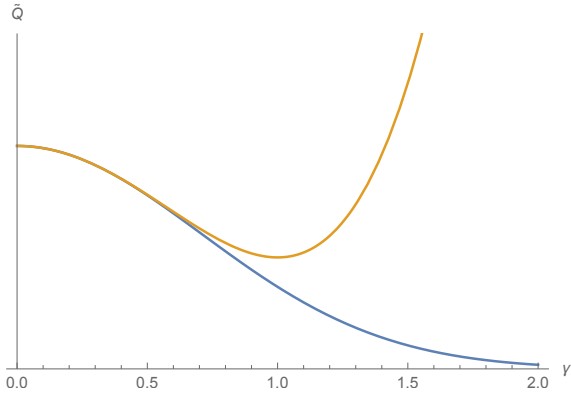

Figure 1: In blue is the difference inequality, (50), which displays exponential damping in $\gamma^2$. In orange is the absolute inequality, (115), which increases (i.e. becomes weaker) for large masses. This plot was made using square-root Gaussian smearing functions, (8), $g(x^+, x^-) = \frac{1}{\sqrt{\delta^+ \delta^-}} f_{\sqrt{G}}(x^+/\delta^+) f_{\sqrt{G}}(x^-/\delta^-)$ and choosing the boost parameter $e^\eta = \sqrt{\delta^-/\delta^+}$ to scale out the dependence on the smearing parameters as outlined in the paragraph above equation (52).

and plot the two bounds as functions of $\gamma$. The two plots are shown in figure 1.

## 5 Boost optimization of the bound

Having derived the DSNEC bounds in section 3.2 and section 4.3 we now address the issue of Lorentz covariance, namely that our boundary involves a explicit reference frame. We remind the reader that is related to a freedom in the choice of smooth sampling domain that we discussed earlier.

In this section we utilize this freedom (or at least a portion of it) optimize the derived bound over Lorentz boosts of the domain. We show in principle how this works in four spacetime dimensions where we derive an "boost-optimized" bound. More generally, however, we show how a simple (albeit sub-optimal) Lorentz covariant bound can be derived in general dimensions. For even spacetime dimensions, this bound can be expressed directly in position space.

### 5.1 4-dimensions

We start with the expression Eq. (109) in $n = 4$ dimensions. Because the integrand is even[8] about $k_\eta \to -k_\eta$ we can extend $D_\eta$ to the entire $\mathbb{R}^2$ plane at the expense of a factor of 1/2:

$$\int d^2x^\pm g(x^+, x^-)^2 \langle T^{\text{ren}}_{--} \rangle_\psi \geq -\frac{1}{12\pi^2} \int \frac{dk_+ dk_-}{(2\pi)^2} \quad |\tilde{g}(k_+, k_-)|^2 \Big( \beta^3 k_+^4 + 4\beta^2 k_+^3 k_-$$
$$+ 6\beta k_+^2 k_-^2 + 4 k_+ k_-^3 + \beta^{-1} k_-^4 \Big), \quad (118)$$

where $\beta \equiv e^{2\eta}$. We will proceed by assuming that the smearing function factorizes as

$$|\tilde{g}(k_+, k_-)|^2 = |\tilde{g}_+(k_+)|^2 |\tilde{g}_-(k_-)|^2, \quad (119)$$

---

[8]Since $g$ is real in position space $\tilde{g}(-k_+, -k_-) = \tilde{g}(k_+, k_-)^*$.

normalized to

$$\int \frac{dk_\pm}{2\pi} |\tilde{g}_\pm(k_\pm)|^2 = 1. \tag{120}$$

We will simplify the notation in what follows by defining moments

$$\int \frac{dk_\pm}{2\pi} k_\pm^n |\tilde{g}_\pm(k_\pm)|^2 \equiv \langle k_\pm^n \rangle, \qquad n \in \mathbb{Z}. \tag{121}$$

While hidden in this notation, it is important to keep in mind that $\langle k_\pm^n \rangle$ is a functional of $\tilde{g}_\pm$, respectively. Additionally note that the assumption that the smearing function factorizes forces the odd moments to vanish. So we have

$$\int d^2 x^\pm g(x^+, x^-)^2 \langle T_{--}^{\text{ren}} \rangle_\psi \geq -\frac{1}{12\pi^2} \left( \beta^3 \langle k_+^4 \rangle + 6\beta^1 \langle k_+^2 \rangle \langle k_-^2 \rangle + \beta^{-1} \langle k_-^4 \rangle \right) \equiv -\mathcal{Q}(\beta). \tag{122}$$

The minimizer of the bound, $\beta_0$, is a real positive solution to

$$\mathcal{Q}'(\beta_0) = 0 \quad \Rightarrow \quad \langle k_-^4 \rangle - 6\beta_0^2 \langle k_+^2 \rangle \langle k_-^2 \rangle - 3\beta_0^4 \langle k_+^4 \rangle = 0, \tag{123}$$

and the boost-optimized bound is

$$\mathcal{Q}(\beta_0) = \frac{\sqrt{\langle k_-^4 \rangle}}{3\pi^2} \frac{\left( \langle k_-^4 \rangle \langle k_+^4 \rangle + 3\langle k_-^2 \rangle \langle k_+^2 \rangle \left( 3\langle k_-^2 \rangle \langle k_+^2 \rangle + \sqrt{9\langle k_-^2 \rangle^2 \langle k_+^2 \rangle^2 + 3\langle k_-^4 \rangle \langle k_+^4 \rangle} \right) \right)}{\left( 3\langle k_-^2 \rangle \langle k_+^2 \rangle + \sqrt{9\langle k_-^2 \rangle^2 \langle k_+^2 \rangle^2 + 3\langle k_-^4 \rangle \langle k_+^4 \rangle} \right)^{3/2}}. \tag{124}$$

Before moving on, let us remark on some features of the above bound. Firstly, we emphasize that the optimization over boosts is only a one-parameter characterization of the freedom in choosing a smooth sampling domain. Thus we strongly suspect that (124) is not the *truly optimal* bound for $4d$ massless scalars.

Secondly, boost optimization has restored Lorentz covariance to (124): under $(k_+, k_-) \to (\lambda k_+, \lambda k_-)$ in the integrals defining the moments of $|\tilde{g}|^2$, $Q \to \lambda^4 Q$ consistent with the engineering dimension of $T_{--}$ and under $(k_+, k_-) \to (\lambda k_+, \lambda^{-1} k_-)$, $Q \to \lambda^{-2} Q$ consistent with the weight of $T_{--}$ under boosts. We will find this to also be true of the bounds we derive for general dimensions.

Thirdly, unlike what is suggested prima facie by (109), (124) is not a linear functional of the original smearing function $g^2$. This follows from solving $Q'(\beta_0) = 0$ in terms of the moments of $|\tilde{g}|^2$, i.e. *the optimizing boost parameter depends on the smearing function*. This is a common feature of the bounds we discuss for general dimensions below.

Lastly, for general dimensions (and in particular odd dimensions) the extension of $D_\eta \to \mathbb{R}^{1,1}$ is only valid for the absolute value of $k_\eta^n$. As a consequence we will not be able to drop odd moments of $|\tilde{g}|^2$. Additionally, solving the resulting polynomial $\mathcal{Q}'(\beta_0) = 0$ may not be analytically possible in generic dimensions. As we will soon see we can circumvent these difficulties and derive a generic expression at the expense of making the bound slightly weaker.

## 5.2 General dimensions

We start by noting

$$\int \frac{d^2 k_\pm}{(2\pi)^2} |\tilde{g}(k_+, k_-)|^2 k_\eta^n \Theta(k_\eta) = \frac{1}{2} \int \frac{d^2 k_\pm}{(2\pi)^2} |\tilde{g}(k_+, k_-)|^2 |k_\eta|^n. \tag{125}$$

We will continue to assume that the smearing function factorizes as in Eq. (119) and normalized as in Eq. (120), and we write

$$\int d^2x^\pm g(x^+, x^-)^2 \langle T_{--}^{\text{ren}} \rangle_\psi \geq -\frac{c_{T_{--}}^{(n)}}{2} \sum_{m=0}^{n} \binom{n}{m} e^{(2+2m-n)\eta} \langle |k_+|^m \rangle \langle |k_-|^{n-m} \rangle, \tag{126}$$

where we recall the definition of $c_{T_{--}}^{(n)}$ in equation (51). Note that we have made use of the triangle inequality after expanding $|k_\eta|^n$ and so we have already weakened the bound. Next we implement Hölder's inequality on each term of the sum. The inequality is the following [35]: given a probability measure $d\mu$ and two measurable functions $f_1$ and $f_2$ then

$$\int d\mu |f_1||f_2| \leq \left( \int d\mu |f_1|^{p_1} \right)^{1/p_1} \left( \int d\mu |f_2|^{p_2} \right)^{1/p_2}, \quad p_{1,2} \geq 1, \quad 1/p_1 + 1/p_2 = 1. \tag{127}$$

For $\langle |k_+|^m \rangle$, for example, we use the measure $d\mu = \frac{dk_+}{2\pi} |\tilde{g}_+|^2$, functions $f_1 = |k_+|^m$, $f_2 = 1$, and $p_1 = n/m$, $p_2 = \frac{n}{n-m}$. This doesn't apply for the $m = n$ term in the sum, however we don't need to implement the inequality for this term. We now have

$$\int d^2x^\pm g(x^+, x^-)^2 \langle T_{--}^{\text{ren}} \rangle_\psi \geq -\frac{c_{T_{--}}^{(n)}}{2} \sum_{m=0}^{n} \binom{n}{m} e^{(2+2m-n)\eta} \langle |k_+|^n \rangle^{m/n} \langle |k_-|^n \rangle^{1-m/n}. \tag{128}$$

Now defining $\bar{\beta} := e^{-\eta} \langle |k_+|^n \rangle^{-\frac{1}{2n}} \langle |k_-|^n \rangle^{\frac{1}{2n}}$ to arrive at

$$\boxed{\int d^2x^\pm g(x^+, x^-)^2 \langle T_{--}^{\text{ren}} \rangle_\psi \geq -\frac{c_{T_{--}}^{(n)}}{2} P_n(\bar{\beta}) \langle |k_+|^n \rangle^{\frac{n-2}{2n}} \langle |k_-|^n \rangle^{\frac{n+2}{2n}},} \tag{129}$$

where we recall the notation $\langle |k_\pm|^p \rangle := \int dk_\pm |\tilde{g}_\pm(k_\pm)|^2 |k_\pm|^p$. All residual dependence on the boost parameter lies in

$$P_n(\bar{\beta}) = \sum_{m=0}^{n} \binom{n}{m} \bar{\beta}^{n-2-2m} = \bar{\beta}^{-2}(\bar{\beta}^{-1} + \bar{\beta})^n, \tag{130}$$

which a simple polynomial of $\bar{\beta}$ and whose optimal value is given by

$$\bar{\beta}_0 = \sqrt{\frac{n+2}{n-2}}, \qquad P_n(\bar{\beta}_0) = \frac{n-2}{n+2} \left( \sqrt{\frac{n-2}{n+2}} + \sqrt{\frac{n+2}{n-2}} \right)^n. \tag{131}$$

Note that much like the $4d$ boost-optimal bound, (124), the general bound (129) has restored covariance under boosts and rescalings of $(k_+, k_-)$ and so fixes the dependence on smearing lengths into the DSNEC form.

For instance, being explicit, we could choose the functions $g_\pm$ as

$$g_\pm(x^\pm) = \frac{1}{\sqrt{\delta^\pm}} f_{\sqrt{G}}(x^\pm / \delta^\pm), \tag{132}$$

where we recall that $f_{\sqrt{G}}$ is the square-root Gaussian, (8). The moments are[9]

$$\langle |k_\pm|^m \rangle = \frac{\Gamma\left(\frac{m+1}{2}\right)}{2^{\frac{m+1}{2}} \sqrt{2\pi} (\delta^\pm)^m}. \tag{133}$$

---

[9]We will choose the convention $\tilde{g}_\pm := \frac{1}{\sqrt{2}} \int dx^+ e^{ik_\pm x^\pm} g_\pm(x^\pm)$ in accordance with the factor of two encountered in footnote 5.

Then we see that Eq. (129) is precisely in the DSNEC form

$$\int \frac{d^2x^\pm}{\delta^+\delta^-} f_{\sqrt{G}}(x^+/\delta^+)^2 f_{\sqrt{G}}(x^-/\delta^-)^2 \langle T^{\text{ren}}_{--}\rangle_\psi \geq -A_n \frac{1}{(\delta^+)^{n/2-1}(\delta^-)^{n/2+1}}\,, \tag{134}$$

where

$$A_n = \frac{c^{(n)}_{T_{--}} P_n(\bar{\beta}_0)\Gamma\left(\frac{n+1}{2}\right)}{2^{\frac{n+3}{2}}\sqrt{2\pi}} = \frac{1}{2(8\pi)^{n/2}}\frac{n-2}{n+2}\left(\sqrt{\frac{n-2}{n+2}}+\sqrt{\frac{n+2}{n-2}}\right)^n\,. \tag{135}$$

It is also helpful to compare this general bound to the "boost-optimal" bound, (124) in four dimensions, again using square-root Gaussians as the smearing functions. A simple calculation reveals that (124) and (129) imply, respectively,

$$\int \frac{d^2x^\pm}{\delta^+\delta^-} f_{\sqrt{G}}(x^+/\delta^+)^2 f_{\sqrt{G}}(x^-/\delta^-)^2 \langle T^{\text{ren}}_{--}\rangle_\psi \; \gtrapprox \; -\frac{2.18\times 10^{-3}}{(\delta^+)^{3/2}(\delta^-)^{5/2}}\,, \qquad \text{(Boost-optimized)}$$

$$\int \frac{d^2x^\pm}{\delta^+\delta^-} f_{\sqrt{G}}(x^+/\delta^+)^2 f_{\sqrt{G}}(x^-/\delta^-)^2 \langle T^{\text{ren}}_{--}\rangle_\psi \; \gtrapprox \; -\frac{7.51\times 10^{-3}}{(\delta^+)^{3/2}(\delta^-)^{5/2}}\,, \qquad \text{(Equation (129))}$$

$$\tag{136}$$

and so indeed (129) is a weaker bound.

## 5.3  Even dimensions

We finish this section of the paper by noting that while generally our bound is most conveniently expressed in momentum space, in even dimensions the integrals can be inverse Fourier transformed to integrals over local quantities in position space (this is not true in odd dimensions because of our bound makes use of the absolute value of the momenta). Indeed by noting

$$\langle |k_\pm|^n\rangle = \frac{1}{2}\int \frac{dk_\pm}{2\pi} |\tilde{g}_\pm|^2 k_\pm^n = \frac{1}{4}\int dx^\pm (g^{(n/2)}(x^\pm))^2\,, \tag{137}$$

where the superscript $(\cdot)^{(n/2)}$ indicates the $n/2$-th derivative, we arrive at simple position space integrals

$$\int d^2x^\pm g(x^\pm)^2 \langle T^{\text{ren}}_{--}\rangle_\psi \geq -\frac{c^{(n)}_{T_{--}} P_n(\bar{\beta}_0)}{8}$$
$$\times \left(\int dx^+ (g_+^{(n/2)}(x^+))^2\right)^{\frac{n-2}{2n}}\left(\int dx^- (g_-^{(n/2)}(x^-))^2\right)^{\frac{n+2}{2n}}\,, \tag{138}$$

where we recall the definitions of the constant $c^{(n)}_{T_{--}}$ in equation (51) and the coefficient $P_n(\bar{\beta}_0)$ in equation (131). In four dimensions, in particular, this is

$$\int d^2x^\pm g(x^+, x^-)^2 \langle T^{\text{ren}}_{--}\rangle_\psi \geq -\frac{16}{81\pi^2}\left(\int dx^+ (g_+''(x^+))^2\right)^{1/4}\left(\int dx^- (g_-''(x^-))^2\right)^{3/4}\,. \tag{139}$$

# 6  Discussion

In this work we investigated the double smeared null energy condition (DSNEC), a proposed bound on the renormalized null energy smeared over two null directions. For free fields in Minkowski space we derived this bound in two separate ways. First, we derived the DSNEC as a

quantum difference inequality using the Minkowski vacuum as the reference state. We showed that this derivation generalizes to bounds on a large set of operators including higher-spin currents. Second, we showed that the DSNEC arises naturally from a general absolute quantum worldvolume inequality. The formalism of this second derivation allows for a straightforward generalization to curved spacetimes. As both approaches require a fixed domain of momentum integration, we further utilized this degree of freedom to optimize the bound over a set of boosted domains. This results in a bound that restores Lorentz covariance and displays an unexpected, non-linear dependence on the smearing function. Finally, we showed how the averaged null energy condition (ANEC) and the smeared null energy condition (SNEC) can be derived from DSNEC at the correct limit.

There are several interesting directions for future work. The most obvious is to investigate the generalization of our bound to curved spacetimes. As mentioned above, this is indeed a primary motivation for expressing the DSNEC as an absolute inequality: as opposed to Minkowski space, there is no preferred reference vacuum state in curved spaces. Renormalizing with respect to the Hadamard parametrix provides a canonical way to derive the DSNEC while allowing for curvature contributions. Generalizing the DSNEC to curved spaces is also a chief concern for applications to semiclassical gravity, which we will return to discuss shortly. Thus this is a direction of high interest and importance.

Further probing the validity of the DSNEC, one can also speculate about its application in generic quantum field theories in Minkowski spacetime. For theories that are relevant perturbations away from free field theory, we generally expect the DSNEC to hold following the argument given in [1]: at large momenta (compared to any inverse correlation lengths of the theory) the divergences appearing in vacuum expectation values are roughly given by those of the UV fixed point. Having shown that the DSNEC holds for the free fixed point, it is reasonable to assume that the smeared null energy is lower bounded in the above situations as well. The engineering dimension of $T_{--}$ and covariance under boosts then fix the schematic form of this bound. For strongly interacting theories, the question becomes more subtle and a proof of the DSNEC will likely require formal CFT techniques. It has been suggested that by looking at states prepared by stress-tensor insertions [36], that energy densities in 4d CFTs obey worldvolume inequalities. It would be interesting to explore whether the same states suggest the validity of the DSNEC and more generally, if the DSNEC can be proven in CFTs. For non-conformal general interacting QFTs, the situation is more difficult as no such QEIs have been derived. Their existence has been established only for operators arising from the operator product expansion of theories satisfying a microscopic phase space condition [37]. However, it is not clear if these operators include components of the stress energy tensor of interacting theories.

Another potential avenue for establishing the validity of the DSNEC in interacting quantum field theories is to explore its connection (if any exists) to state-dependent entropic bounds. The most pressing example is the QNEC, which, as discussed in the introduction, has been proven to hold in generic Poincaré-invariant quantum field theories. There is a broad expectation that the QNEC is tight for interacting theories: for holographic CFTs it is known that the QNEC is saturated [38] and there are strong arguments (although not a strict proof) for QNEC saturation in interacting CFTs [39] (these arguments break down precisely for free theories). Speculating along similar lines, one may hope that any link between the DSNEC and the generalized second law, or the quantum focusing conjecture may more directly establish a role in constraining semi-classical gravity. However, it is not presently clear to us how one can effectively remove the non-linear state-dependence of the QNEC through "double smearing" and arrive at a finite state-independent bound such as the DSNEC. We finally remark on the possibility of proving the DSNEC in the realm of holographic CFTs. Using the principle of "no bulk shortcut" applied to boundary null geodesics, Leichenauer and Levine were able

to prove the SNEC [40] for such theories. For the DSNEC one would like to extend this reasoning to a diamond in the $(x^+, x^-)$ plane. One strategy is foliating the smearing region with null-geodesics and applying "no bulk shortcut" to each geodesic, although some care will be needed in not arriving simply at the SNEC trivially integrated over the other null direction. Perhaps the broader principle of "causal wedge nesting" [41] is a more natural starting point for the DSNEC; more ambitiously, a proof utilizing "entanglement wedge nesting" [42] might also point towards connections with the QNEC (which follows from entanglement wedge nesting in holographic theories [43]). We leave all of these points as interesting and open lines for future research.

The other main direction for future work is to use this type of bound in order to prove singularity theorems. Penrose showed, assuming the NEC, that trapped surfaces lead to singularities [21]. These theorems are violated in semi-classical gravity due to NEC violation. Thus, a result like our bound gives the natural starting point (replacing the NEC) for proving semi-classical singularity theorems. One main obstacle for the DSNEC as an assumption is that singularity theorems require bounds on individual null geodesics. In Appendix B we explore one "light-ray limit" of the DSNEC in showing how to reproduce the SNEC. The SNEC has been used as an assumption to a semiclassical singularity theorem for null geodesic incompleteness [16]. However in the context of field theory alone, the SNEC bound is not very useful as one must then make sense of the UV cutoff. An alternative direction is using the DNEC along with "segment inequality" theorems [44]. Such theorems use worldvolume bounds on the Ricci tensor to show that the length of any geodesic maximizing the distance to a Cauchy surface is bounded, thus establishing singularity theorems.

# Acknowledgements

We thank Tarek Anous, Mert Besken, Chris Fewster and Dimitrios Krommydas for helpful conversations.

**Funding information** JRF is supported the ERC Starting Grant GenGeoHolo. BF and E-AK are supported by the ERC Consolidator Grant QUANTIVIOL. This work is part of the $\Delta$-ITP consortium, a program of the NWO that is funded by the Dutch Ministry of Education, Culture and Science (OCW).

# A On the validity of the Assumption 1

In this appendix we derive the constraints on what types of operators satisfy Assumption 1 from section 3. In line with the general philosophy that a QFT is defined by its flow away from a conformal fixed point (and to be concrete), let us spell out what Assumption 1 implies for primary operators in a CFT (if Assumption 1 is satisfied by a primary then it is also satisfied by its descendants by acting with derivatives). Thus the main object of interest is

$$[\mathcal{O}_\Delta(t, \vec{x}), \mathcal{O}_\Delta(0)], \tag{A.1}$$

where $\Delta$ is the conformal weight of $\mathcal{O}_\Delta$. For simplicity we will focus scalar primaries although our main conclusions will be unchanged for spinning operators. This can be evaluated with

an appropriate $i\varepsilon$ prescription and using the operator product expansion (OPE) [45]

$$[\mathcal{O}_\Delta(t,\vec{x}),\mathcal{O}_\Delta(0)] = \lim_{\varepsilon\to 0}\{\mathcal{O}_\Delta(t-i\varepsilon,\vec{x})\mathcal{O}_\Delta(0) - \mathcal{O}_\Delta(t+i\varepsilon,\vec{x})\mathcal{O}_\Delta(0)\}$$

$$= \lim_{\varepsilon\to 0}\sum_{\Delta'}\mathcal{G}^{\Delta'}_{\Delta\Delta}(\bar{t},\vec{x};\partial)\Big|_{\bar{t}=t+i\varepsilon}^{\bar{t}=t-i\varepsilon}\mathcal{O}_{\Delta'}(0),\tag{A.2}$$

where the sum goes over all other primaries $\Delta'$ and we intend this as an operator statement true inside all Wightman functions with other local operators; via the operator-state correspondence this statement holds in a dense set of the Hilbert space. The identity, $\hat{1}$, (with $\Delta'=0$) *always* appears in the operator product expansion of $\mathcal{O}_\Delta$ with itself. We are interested in what other primaries can possibly contribute to this commutator. To isolate the contribution of a primary $\mathcal{O}_{\Delta'}$ we can consider the overlap of (A.2) in the conformal vacuum, $\Omega_c$, with $\mathcal{O}_{\Delta'}(y)$ in the limit that $|y|\to\infty$. For instance a typical term is

$$\lim_{|y|\to\infty}\langle\mathcal{O}_{\Delta'}(y)\mathcal{O}_\Delta(x)\mathcal{O}_\Delta(0)\rangle_{\Omega_c} = \lim_{|y|\to\infty}\mathcal{G}^{\Delta'}_{\Delta\Delta}(x;-\partial^{(y)})\frac{1}{|y|^{2\Delta'}},\tag{A.3}$$

where we have used the universal form of primary two-point functions. In the $|y|\to\infty$ limit the contribution of descendants (coming from acting by $\partial^{(y)}$) are subleading and so the leading contribution comes the primary operator itself. We can also evaluate the left-hand side of (A.3) using the universal form of conformal three-point function and find

$$\mathcal{G}^{\Delta'}_{\Delta\Delta}(x;0) = \lim_{|y|\to\infty}|y|^{2\Delta'}\langle\mathcal{O}_{\Delta'}(y)\mathcal{O}_\Delta(x)\mathcal{O}_\Delta(0)\rangle_{\Omega_c} = \lim_{|y|\to\infty}\frac{|y|^{2\Delta'}c^{\Delta'}_{\Delta\Delta}}{|y-x|^{\Delta'}|x|^{2\Delta-\Delta'}|y|^{\Delta'}}$$

$$= \frac{c^{\Delta'}_{\Delta\Delta}}{|x|^{2\Delta-\Delta'}}.\tag{A.4}$$

Here $c^{\Delta'}_{\Delta\Delta}$ are three-point coefficients and are c-numbers specifying the CFT and its operator content. To take stock, the contribution of primary operators to the commutator is then

$$[\mathcal{O}_\Delta(t,\vec{x}),\mathcal{O}_\Delta(0)] \supset \sum_{\Delta'}c^{\Delta'}_{\Delta\Delta}\mathcal{O}_{\Delta'}(0)f_{\Delta-\frac{1}{2}\Delta'}(t,\vec{x}),\tag{A.5}$$

with

$$f_h(t,\vec{x}) := \lim_{\varepsilon\to 0}\left\{\frac{1}{(-(t-i\varepsilon)^2+\vec{x}^2)^h} - \frac{1}{(-(t+i\varepsilon)^2+\vec{x}^2)^h}\right\}.\tag{A.6}$$

Note that if a primary $\mathcal{O}_{\Delta'}$ appears in the OPE (i.e. $c^{\Delta'}_{\Delta\Delta}\neq 0$), it is still possible to vanish in the commutator as long as the associated $f_{\Delta-\Delta'/2}$ vanishes as a distribution,

$$\int dt\,\varphi(t)f_h(t,\vec{x})=0 \qquad \forall\text{ smooth }\varphi(t).\tag{A.7}$$

This requires $h\in\mathbb{Z}_{\leq 0}$. For instance if $h\in\mathbb{Z}_{>0}$ then the poles in $f_h$ can contribute to the integration against a test function and so as a distribution [45]

$$f_{h\in\mathbb{Z}_{>0}}(t,\vec{x}) = \frac{2\pi i}{\Gamma(h)}\left\{(t-|\vec{x}|)^{-h}\partial_t^{h-1}\delta(t+|\vec{x}|) + (t+|\vec{x}|)^{-h}\partial_t^{h-1}\delta(t-|\vec{x}|)\right\}.\tag{A.8}$$

And if $h\notin\mathbb{Z}$, $f_h$ possesses branch cuts contributing to integrations against test functions leading to [45]

$$f_{h\notin\mathbb{Z}} \propto \sin(\pi h)(t^2-\vec{x}^2)^{-h}\left\{\Theta(t-|\vec{x}|)-\Theta(t+|\vec{x}|)\right\}.\tag{A.9}$$

Thus we are lead to conclude that as a necessary condition for only the identity operator to appear in commutator $[\mathcal{O}_\Delta, \mathcal{O}_\Delta]$, the only primaries that can appear in $\mathcal{O}_\Delta \mathcal{O}_\Delta$ OPE have conformal dimensions

$$\Delta' = 2\Delta + m \qquad m \in \mathbb{Z}_{\geq 0}. \tag{A.10}$$

Since descendent operators have conformal dimensions differing from primaries by positive integers this is also a sufficient condition. This is very constraining of the operator spectrum of a CFT. We in fact already know one set of primaries that naturally in appear in such an OPE which are the so-called "double-trace" operators schematically of the form

$$\mathcal{O}_{\Delta'}(x) \sim\, : \mathcal{O}_\Delta (\partial^2)^\ell \partial^{\mu_1} \ldots \partial^{\mu_s} \mathcal{O}_\Delta : (x) \qquad \Delta' = 2\Delta + 2\ell + s. \tag{A.11}$$

Excepting the possibility of multiple conformal modules possessing the same conformal weight, we find that $\mathcal{O}_\Delta$ only has its double-traces in its OPE which implies that higher-point functions follow from Wick contractions, reminiscent of free fields. We make the passing remark that the bounds we derive in section 3 make no use of the particular form of commutator, only that it is proportional to the identity and so we make no requirements on the conformal dimension of $\mathcal{O}_\Delta$ itself. This allows for the possibility for $\mathcal{O}_\Delta$ to be a *generalized free field* at some interacting fixed point with some large parameter $N$ suppressing non-Wick contractions by powers of $1/N$. Thus our construction in section 3 provides lower bounds on such primaries (and their descendants) to leading order in $1/N$. However to re-iterate our warning from section 3 these operators in generalized free field theories bounded by our method do not include the stress-tensor unless the field is strictly free (since the stress-tensor is typically not quadratic in the generalized free field unless that field is free).

Moving away from the fixed point we can ask how deformations of the CFT affect the above analysis. For one, we expect that in order to preserve the equal-time commutators, $[\mathcal{O}_\Delta(0, \vec{x}), \mathcal{O}_\Delta(0)]$, in Assumption 1 irrelevant deformations should be prohibited as they might contain derivative couplings (or perhaps might induce derivative couplings via renormalization) that can alter the canonical structure. However, we also require Assumption 1 to hold over all points in a causal domain and so we must evolve $\mathcal{O}_\Delta(0, \vec{x})$ away from $t = 0$ using the interacting Hamiltonian. This will generically generate more operators unless the deformation is Gaussian. Given these two arguments it seems that Assumption 1 will only hold for the simplest massive deformation of a generalized free field fixed point: $m^2 \mathcal{O}_\Delta \mathcal{O}_\Delta$.

## B    SNEC from DSNEC

It is clear that in a "light-ray" limit, say, by taking the $\delta^+ \to 0$, the right-hand side of the DSNEC diverges leaving a trivial bound. This is expected on general grounds: the null-energy averaged along a finite portion of a light-ray is unbounded from below in QFT [17]. However, with the introduction of a UV cutoff of the theory, the bound remains finite allowing the proof of SNEC [1,2]. Here, we investigate the derivation of the "field theory" version of SNEC from DNEC at the appropriate limit.

First we examine the schematic form of DNEC (52), imposing the following cutoff: we take $\delta^+ \to 0$ while $\delta^+ \delta^- \to \ell_{UV}^2$. Similarly to the way we derived ANEC we require that the smearing function factorizes and $\lim_{\delta^+ 0} f_+(x^+/\delta^+)^2/\delta_+ = \delta(x^+ - \beta)$. Then we have

$$\int dx^- g_-(x^-)^2 \langle T_{--}(x^+ = \beta, x^-) \rangle_\Psi \geq -\frac{\mathcal{N}_2}{\ell_{UV}^{n-2}(\delta^-)^2}, \tag{B.1}$$

consistent with a schematic form of the SNEC.

To investigate if the DNSEC implies a SNEC type bound with the same number of derivatives on the smearing function we start from Eq. (108). Picking the $\eta \to -\infty$ limit of our boosted domains $D_\eta$ the equation becomes[10]

$$\int d^2x^\pm g(x^\pm)^2 \langle T^{\text{ren}}_{--} \rangle_\psi \gtrsim -\int d^2k_\pm |\tilde{g}(k_+,k_-)|^2 \int_0^{k_-} d\zeta_- \int_0^\infty d\zeta_+ (\zeta_+)^{n/2-2}(\zeta_-)^{n/2}\Theta(k_-). \tag{B.2}$$

Then the right-hand side is independent of $k_+$. One might now be tempted to take the $\delta^+ \to 0$ limit on both sides. However we have only moved the divergence to a new place: the unbounded $\zeta_+$ integration. We cannot infinitely boost $D_\eta$ and expect a finite lower bound (indeed in this limit $D_\eta$ fails to satisfy the criteria of a small sampling domain). This is the point at which we implement the UV cutoff. We place this cutoff covariantly on $\zeta_\pm$

$$\zeta_+ \zeta_- < \ell_{UV}^{-2}. \tag{B.3}$$

In our first approach we additionally assume that momenta appearing in the state are cutoff as $\zeta_\pm \leq \Lambda_\pm$ This is no longer a Lorentz invariant cutoff as imposed in some versions of SNEC. However, this approach will allow us to derive SNEC for momenta arbitrarily close to $\Lambda_\pm$. Then the $\zeta_+$ integral is bounded and Eq. (B.2) becomes

$$\int d^2x^\pm g(x^\pm)^2 \langle T^{\text{ren}}_{--} \rangle_\psi \gtrsim -\int d^2k_\pm |\tilde{g}(k_+,k_-)|^2 \Lambda_+^{n/2-1} k_-^{n/2+1}. \tag{B.4}$$

We will assume that the support of the smearing function in momentum space, $|\tilde{g}|^2$, only has support for momenta below these cutoffs, $k_\pm \leq \Lambda_\pm$

$$\int d^2x^\pm g(x^\pm)^2 \langle T^{\text{ren}}_{--} \rangle_\psi \gtrsim -\frac{1}{\ell_{UV}^{n-2}} \int \frac{d^2k_\pm}{(2\pi)^2} |\tilde{g}(k_+,k_-)|^2 k_-^2 \gtrsim -\frac{1}{\ell_{UV}^{n-2}} \int d^2x^\pm (\partial_- g(x^\pm))^2. \tag{B.5}$$

Evaluating the stress-tensor on the $x^-$ direction and assuming that the smearing function factorizes as

$$g^2(x^\pm) = g_-^2(x^-) g_+^2(x^+), \tag{B.6}$$

reproduces the SNEC bound

$$\int dx^- g_-(x^-)^2 \langle T^{\text{ren}}_{--} \rangle_\psi \geq -\frac{1}{\ell_{UV}^{n-2}} \int dx^- (\partial_- g_-(x^-))^2. \tag{B.7}$$

In a different approach, we do not independently bound the $\zeta_\pm$ momenta but just implement the covariant cutoff of Eq. (B.3). On mass-shell, $4\zeta_+\zeta_- = \vec{\zeta}_\perp^2 + m^2$, so one can view this as a cutoff on the transverse momenta accessible to the theory. This is the same regime in which light-sheets admit a "pencil decomposition" and in which the SNEC was proven in [1]. Revisiting (108) in the large $e^{-\eta}$ limit, the upper limit of the $\zeta_+$ integration is approximately replaced with

$$e^{-2\eta}(\zeta_- - k_-) \leq \zeta_+ - k_+ < \frac{1}{\ell_{UV}^2 \zeta_-}(1 - \ell_{UV}^2 k_+ \zeta_-). \tag{B.8}$$

If we additionally assume that the maximum momenta for which the smearing function has support obeys $(k_+)_{max}(k_-)_{max} \ll \ell_{UV}^{-2}$, then the second term of (B.8) is subleading (recall that

---

[10]For the rest of this calculation we will ignore constant prefactors of order one for simplicity.

$\zeta_-$ integral only has support for $\zeta_- < k_-$). Implementing this back into (108)

$$\int d^2x^\pm g(x^\pm)^2 \langle T_{--}^{\text{ren}} \rangle_\psi \gtrsim$$

$$-\int \frac{d^2k_\pm}{(2\pi)^2} |\tilde{g}(k_+, k_-)|^2 \int_0^{k_-} d\zeta_- \int_0^{\frac{1}{\ell_{UV}^2 \zeta_-}} d\zeta_+ (\zeta_+)^{n/2-2} (\zeta_-)^{n/2} \Theta(k_-)$$

$$\gtrsim -\frac{1}{\ell_{UV}^{n-2}} \int \frac{d^2k_\pm}{(2\pi)^2} |\tilde{g}(k_+, k_-)|^2 k_-^2 \gtrsim -\frac{1}{\ell_{UV}^{n-2}} \int d^2x^\pm (\partial_- g(x^\pm))^2, \quad \text{(B.9)}$$

again arriving at the SNEC as before. Note that in this covariant approach the assumption that $(k_+)_{max}(k_-)_{max} \ll \ell_{UV}^{-2}$ places a strong limitation on how finely one can probe the light-ray, depending on the UV cutoff.

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
