# Peer review of "The double smeared null energy condition"

_SciPost Physics, doi:SciPost Phys. 14, 024 (2023)_

## Round 1 · Referee Report · Anonymous (Referee 1) · 2022-3-10

Report

The authors propose a new bound on the stress tensor of quantum fields in gen- eral spacetimes, known as the Double Smeared Null Energy Condition (DSNEC). The stress tensor in question is smeared along null directions and such stress tensor will generally scale with the smearing lengths simply based on symmetry, so the non-trivial result that the authors obtain is a bound on the coefficient multiplying the form obtained through symmetries. The paper’s relevance con- sists of their method of obtaining such a bound, which will allow future work to include curvature corrections to the energy bound (in a state-independent way). However, there are several points which, if clarified correctly, can benefit the reader of the paper greatly. Namely, some technical points include:

•The authors went through the trouble of writing out full equations and many steps in the calculations, but failed to properly define several quan- tities that appear throughout the calculations. For instance, in section 2, the authors mention the Hadamard parametrix, although they do not define what it means, but leave it for section 4, which makes the section 2 unfollowable on its own. Likewise, the equation 16 is supposed to be definitional for what one might mean by a certain expectation value of the stress tensor, but they use a symbol o which is not defined on its own, and so on. Please make sure all of the objects, variables and operations are properly defined throughout the text.

•The optimization over boosts is interesting, although I’m not quite sure I fully understand the logic. Namely, the equation 34 tells us to mini- mize over the domains for the boost parameter. Why is minimization the correct way and not extremization in general? As far as I understand, the point of the optimization is to give you the tightest bound on the stress tensor, so I’m not sure why that is implied by the minimization. The footnote that provides some explanation regarding the optimization procedure is likewise somewhat unclear: I understand that by performing such a procedure, one is obtaining a local minima so to say, and therefore, infinitesimal coordinate changes cannot change the domain and hence, we obtain a coordinate-independent formulation. However, it is not clear that, in this language, this is a global minima of D; could you please clarify this point?

•Under equation 47, there is a sentence: ”Processing h_D a bit, we obtain...”. Please elaborate.

•In the ”HSANEC” paragraph, does the acronym stand for Half-Sided ANEC? Please define the term properly and elaborate on its relevance.

•Please expand on the last sentence on page 15: ”However, Ref. [4]...”, since the sentence by itself does not contain any information.

•Please check equations 120 and 121.

Overall, the paper mostly consists of printing out the relevant calculations, but the reader would benefit from some physical intuition on why some of these bounds are relevant and what is their physical significance. Some conceptual points include:

•Please elaborate on the physical relevance of the quantum worldvolume inequality.

•Could you also elaborate on why AANEC is not ”enough” to consider when it comes to semiclassical (self-consistent) solutions? One of the reasons why AANEC is a natural bound also comes from the fact that it can be derived from the GSL (this was proven by Wall in 0910.5751). Is there any sense in which one can derive the DSNEC in a similar way? If such a derivation is possible, then this new bound can be connected with other known developments in the context of semiclassical gravity (such as the QFC), which would then reaffirm its relevance. It is also not clear how one would independently of AANEC obtain the QNEC bound, given that this one is state-dependent (the authors also acknowledge this).

•It is mentioned that one possible application of this bound would be for proving singularity theorems. However, there have been already several different approaches to the singularity theorems, and especially if they require the non-compactness of the Cauchy slice on which the trapped surface resides, then that significantly weakens the usefulness of the new bound (all one requires in that case is an argument why a geodesic must be incomplete in some sense). If the singularity theorem shows new results, then it would be quite significant: for instance, there was a recent paper 2201.11132 on a new take on singularity theorems which also leads to unexpected results (such as a singularity in an island setting where it wouldn’t be expected). It would be quite interesting to show that such a singularity would be seen through a DSNEC bound. After all, a bound is only as useful as its applications.

•Another point of interest is the proof of DSNEC in non-trivial situations: it is already stated that the provided proof will go through for generalized free fields and large N theories. Have you thought about obtaining a proof through the holographic duality for the strongly-coupled CFTs, that is to say, by looking only at the bulk causal/entanglement features?

Overall, the paper points to some sort of a bound that certainly exists at least for free fields. If it can be used in an interesting way, the bound can pave the way for new results within the regime of semiclassical gravity. Therefore, I recommend the paper to be published, as long as the points mentioned above are properly addressed.

  • validity: -
  • significance: -
  • originality: -
  • clarity: -
  • formatting: -
  • grammar: -

Author:  Eleni-Alexandra Kontou  on 2022-06-02  [id 2551]

(in reply to Report 1 on 2022-03-10)

We thank the referee for reviewing our manuscript in depth and for their suggestions for improving its readability and for alerting us to places where we could be more clear in the explanation of our results.

  1. Regarding the definition of the Hadamard parametrix: we maintain that for section 2, where we discuss our renormalization scheme in generality, our schematic definition of the parametrix as the bidistribution collecting the singularity structure of Hadamard states is sufficient and the details are best left to section 4 where they are utilized (this also does not distract the discussion away from section 3 which is establishes a difference inequality and does not utilize the details of the Hadamard parametrix). In the interest of clarity for the reader we have also added a reference beside the schematic definition and also added a sentence regarding the parametrix for the massless scalar as a heuristic example for the reader. Regarding the referee's more general point, we have added clearer definitions to the "circle operation" (below eq (16)) as the referee suggested, the integration measure in footnote 1, the axioms for the renormalized stress tensor (section 2), and the ultra-regular domain (section 2). We thank the referee for drawing our attention to the need for this.

  2. In general since the object on the right-hand side of equation (34) is negative, if we are searching for the tightest lower bound then minimization is the correct procedure. We acknowledge that, in practice, we perform this minimization as an optimization problem and so it is possible that we find local minima to equation (34) (as opposed to the global minimum). However in section 5.1 we perform the minimization over boosted domains explicitly for four dimensions finding the global minimum (this is guaranteed since the equation we are optimizing is polynomial and we can check each solution explicitly).

  3. We have included an equation (now equation (48)) and a new sentence below it to better explain the processing of $h_D$.

  4. "HSANEC" stands for "Higher Spin ANEC" as adopted from reference [3] where it first appears. Seeing as we had not defined the acronym properly beforehand we changed the heading to "Higher Spin ANEC."

  5. The sentence is now modified and extended to include more information.

  6. Equations (120) and (121) were checked. (120) is correct and remains, while indeed (121) was not correct (it used an older convention that we had neglected to change) and has been modified to be consistent with (120). The solution to the minimization remains unchanged.

  7. The absolute QEI is physically expressing the restrictions on negative energies in quantum field theory. As this is a general feature of QEIs a sentence was added in the introduction.

  8. The AANEC is definitely sufficient for certain applications. However it cannot be used in cases such as singularity theorems where a bound on geodesic segments is required. Regarding relations to QNEC and the GSL speculated by the referee, it is not clear to us presently how to related the DSNEC to state-dependent entropic bounds. Some sentences have been added to the (new) fourth paragraph of the discussion section discussing this.

  9. There is some information in the discussion section on the application of DSNEC for singularity theorems which is the topic of current work. However, replacing pointwise energy conditions with average ones does not change the requirement for non-compactness of the Cauchy surface.

  10. We thank the referee for this comment since firstly it points out an opportunity to improve the clarity of our proof for generalized free fields (and large N theories): in section 3 we prove a lower bound on smeared operators that are squares of generalized free field operators (or derivatives of generalized free fields). For exactly free fields this includes the stress tensor (and proves the DSNEC), however for generalized free fields this does not include the stress tensor. A sentence in the paragraph underneath equation (30) as well as a sentence in the paragraph under equation (A.11) have been added to point this out to the reader. Secondly, because of the above point, proving DSNEC for holographic CFTs (which admit a generalized free field structure) still remains a non-trivial task for future research. We have added a paragraph to the discussion (fourth paragraph) speculating on the possibility of holographic proofs of the DSNEC.

---

## Round 1 · Referee Report · Anonymous (Referee 2) · 2022-3-29

Strengths

1- Derivation of central claim is clearly laid out and easy to follow 2- Comparison with previous literature to support central claim and compare similarities and differences are very clearly and explicitly explained throughout the whole paper 3-The DSNEC appears to have applicability for bounding other quantities beyond the energy-momentum tensor

Weaknesses

1-Derivation only for free fields in Minkowski spacetime. Abstract claims "Our method allows for future systematic inclusion of curvature corrections." but this is only briefly mentioned in the discussion 2-Some of the intermediate results lack clear definitions 3- There is some confusion regarding UV divergences at the start of the central derivation 4- Boost optimization involves additional inequalities so the optimal bound is not found

Report

The paper's main objective is to derive the double smeared null energy condition, which bounds how negative can a specific average over two null directions of the energy momentum tensor be. It appears to recover previous results in the literature and allow for generalisation for bounding other quantities. However, the methods laid out seem to be rely somewhat heavily on free fields (albeit generalised free fields) and Minkowski spacetime.

It is very well written and laid out in quite a pedagogical manner. Even readers not entirely familiar with this field ought to have no trouble following the narrative and understanding the central points of the manuscript. However, when following there were a couple of questions left unanswered and it was lacking a few definitions.

1- In footnote 1, despite attempt at clarifying notation it was slightly unclear whether $d^2x^\pm=dt~dx^1=\frac{1}{2}dx^+~dx^-$ or $d^2x^\pm=dx^+~dx^-=2dt~dx^1$. Clarification of this would help readability.

2- In section 2, the definition of ultra-regular domain is left to a reference, this generates some confusion. It would be better for the definition to be explicit in the text. Given it seems that the properties of these domains are not essential to the derivation, a quick intuitive definition should be sufficient.

3- In the paragraph before Eq. (18) exactly the same issue arises but with respect to the definition of $T^{ren}$. Explicit statement of its definition, or an intuitive explanation would help clear the confusion surrounding that paragraph.

4- Equations (26) and (27) are exactly equal, it is merely a re-writing of the same expression. However, the way the text is presented, talking of point-splitting the operators, suggests that in fact they are no longer equal. A remark emphasising that they are still exactly equal would be important to aid the reader.

5- After Eq. (30) it is claimed that the restriction to the half-space makes the point splitting real rather than fictitious, softening contact divergences. However, this does not seem to be correct.

Under Assumption 1 the integrand is symmetric so Eq. (30) and (27) are still exactly equal, we can perfectly carry the reasoning backwards to go back to the full space. Therefore if (26) is UV-divergent, then (30) would also have to be UV-divergent. It therefore does not seem correct to state this manipulation softens contact divergences.

6- In a related note, it seems unclear why is (30) a finite quantity. If I have understood section 2 correctly, then the two terms in (27) are separately infinite and it is only their difference which is finite. This would suggest (30) is also infinite. Later on in the paper (118) appears to suggest the smearing is taking care of UV-divergences and making both terms in (27) finite. If this is true this should be mentioned at the very start of section 3, right after (26) is written. If this is not true then it is a major flaw in the reasoning in the paper. An alternative argument as to why are both terms in (27) finite would then need to be provided.

7- Minor typo: before Eq. (40) it should read $Q_{T_{--}}[D_\eta]$ rather than $Q_{T_{++}}[D_\eta]$

8- Typo in Eq. (71), the domain of integration in the second line is written as $\mathcal{D}$ which is an operator, not a domain. It should be clarified what is the correct domain of integration.

9- Minor typo in the paragraph after Eq. (71) it should read "where $\mathcal{D}$ is a partial differential operator" rather than "where $\mathcal{D}$ is partial differential operator"

10- Minor typo in second to last paragraph of section 5.1, it should read "is not a linear functional" rather than "is not linear functional"

In summary, this is a very relevant paper which paves the way to new avenues of research in semiclassical gravity. There are many immediate follow-up questions one can pursue. If the points raised above are addressed I certainly recommend its publication.

Requested changes

1- Clarification of footnote 1 2- Present definition of ultra-regular domain, at least intuitively 3- Present axioms that define $T^{ren}$, at least intuitively 4- Make it clear that (26) and (27) are exactly equal 5- Remove statement that (30) softens divergences 6- Clarify if the terms in (27) are independently finite 7- Correct the typos presented in main report above 8- Remove statement "Our method allows for future systematic inclusion of curvature corrections." in the abstract or add a section on how to systematically include curvature corrections.

  • validity: good
  • significance: high
  • originality: high
  • clarity: good
  • formatting: excellent
  • grammar: excellent

Author:  Eleni-Alexandra Kontou  on 2022-06-02  [id 2550]

(in reply to Report 2 on 2022-03-29)

We thank the referee for the careful review of our paper and for pointing out important clarifications.

  1. The footnote was modified to make the definition of $d^2x^\pm$ clear.

  2. A definition of the ultra-regular domain was added.

  3. The paragraph is now slightly reorganized to express the meaning of Tren.

  4. A sentence was added below equation (27) to emphasize that it is equal to equation (26) and that indeed the said point-splitting is fictitious.

  5. Equation (30) is not exactly equal to equation (27), as the referee suggests. This is because the integrand of equation (30) is no longer symmetric under the interchange of x and x': the commutator of O(x) and O(x') is not zero. By the key assumption (that the commutator is proportional to identity) it is only the difference in expectation values that is symmetric, and so after dropping the positive part of the integrand we cannot "go backwards to go back to the full space" and recover a delta function. This is also the mechanism for how the "fictitious" point-splitting in equation (27) is softened into a true point splitting. This is exemplified by equation (32) where the delta function has been replaced by i/(t-t'). The paragraph underneath equation (30) has been expanded to explain this further and a sentence under equation (32) has been added to better illustrate the point-splitting mechanism.

  6. The finiteness of equation (30) is explained in the above point. We thank the referee for pointing these two points of confusion out to us since it alerted us to the need to better explain the logic of this section.

7.-10. The typos are now fixed.

Requested changes:

  1. See point 1. above.

  2. See point 2. above.

  3. See point 3. above. Additionally the role of the axioms is explained and some of them are mentioned.

  4. See point 4. above.

  5. See point 5. above.

  6. See point 4. and point 5. above. A sentence stating that both terms of (27) are infinite (while the difference is finite) has been added to the expanded paragraph below equation (30).

  7. The typos are now fixed.

  8. We believe the statement is correct as our method allows for the systematic inclusion of curvature. Even though the explicit computation of curvature is not done in the present work, Eq.(84) is valid for curved spacetimes. The bound can be calculated with methods described in the citation given and the reduction of the integral in the two null directions can follow our analysis for Minkowski space. This is now explained in some detail under Eq.(84).

---

## Round 2 · Referee Report · Anonymous (Referee 1) · 2022-6-5

Report

The authors have properly addressed all of the points raised previously, and therefore, I highly recommend this paper to be published.

---

## Round 2 · List of Changes

-Introduction: Added a sentence about the physical meaning of QEIs.

-Introduction: A reference on relative entropy was added.

-Section 2: Included more details on the renormalization procedure.

-Section 2: Explained the circle operator and the ultra regular domain.

-Section 2: The connection between the normal ordered stress energy tensor and the renormalized stress energy tensor is explained more.

  • Section 3: The difference between equations (27) and (30) is now more explicit. The point-splitting procedure has additional details.

-Section 4: More details have been added regarding the incorporation of curvature in the bound.

-Section 4: Eq.(121) is fixed as it included wrong signs which however do not affect the rest of the calculation.

-Discussion: A paragraph was added on the topic of proving the DSNEC for holographic QFTs.

  • Appendix B: The SNEC bound that was proven is now written explicitly.

-General: Typos were fixed throughout the manuscript.

---

## Editorial Decision

published